# Deletion of a kinesin I motor unmasks a mechanism of homeostatic branching control by neurotrophin-3

**Thomas O Auer**[1,2,3,4]*[†], **Tong Xiao**[5,6][†], **Valerie Bercier**[1,2,3], **Christoph Gebhardt**[1,2,3], **Karine Duroure**[1,2,3], **Jean-Paul Concordet**[7], **Claire Wyart**[8], **Maximiliano Suster**[9,10], **Koichi Kawakami**[10], **Joachim Wittbrodt**[4], **Herwig Baier**[5,11], **Filippo Del Bene**[1,2,3]*

[1]Institut Curie, Centre de Recherche, Paris, France; [2]CNRS UMR 3215, Paris, France; [3]INSERM U934, Paris, France; [4]Centre for Organismal Studies, University of Heidelberg, Heidelberg, Germany; [5]Department of Physiology, University of California San Francisco, San Francisco, United States; [6]Department of Chemistry, University of California, Berkeley, Berkeley, United States; [7]Muséum National d'Histoire naturelle, Inserm U 1154, CNRS, UMR 7196, Muséum National d'Histoire Naturelle, Paris, France; [8]Institut du Cerveau et de la Moelle épinière, ICM, Inserm U 1127, CNRS, UMR 7225, Sorbonne Universités, UPMC University Paris 6, Paris, France; [9]Neural Circuits and Behaviour Group, Uni Research AS High Technology Centre, Bergen, Norway; [10]Division of Molecular and Developmental Biology, National Institute of Genetics, Shizuoka, Japan; [11]Department Genes–Circuits–Behavior, Max Planck Institute of Neurobiology, Center for Integrated Protein Science Munich (CIPSM), Martinsried, Germany

**\*For correspondence:** thomas. auer@curie.fr (TOA); filippo.del-bene@curie.fr (FDB)

[†]These authors contributed equally to this work

**Competing interests:** The authors declare that no competing interests exist.

**Reviewing editor**: Graeme W Davis, University of California, San Francisco, United States

**Abstract** Development and function of highly polarized cells such as neurons depend on microtubule-associated intracellular transport, but little is known about contributions of specific molecular motors to the establishment of synaptic connections. In this study, we investigated the function of the Kinesin I heavy chain Kif5aa during retinotectal circuit formation in zebrafish. Targeted disruption of Kif5aa does not affect retinal ganglion cell differentiation, and retinal axons reach their topographically correct targets in the tectum, albeit with a delay. In vivo dynamic imaging showed that anterograde transport of mitochondria is impaired, as is synaptic transmission. Strikingly, disruption of presynaptic activity elicits upregulation of Neurotrophin-3 (Ntf3) in postsynaptic tectal cells. This in turn promotes exuberant branching of retinal axons by signaling through the TrkC receptor (Ntrk3). Thus, our study has uncovered an activity-dependent, retrograde signaling pathway that homeostatically controls axonal branching.

## Introduction

Intracellular transport is an essential process in cell growth, maintenance, and inter- and intracellular signaling. This is especially apparent in highly polarized cells like neurons that are composed of complex dendrites and a long axon responsible for impulse propagation. Most of the proteins, mRNAs, and organelles required for cellular growth and function are produced in the cell body and must, therefore, be moved down the axon to the synaptic terminals. Microtubules serve as main longitudinal cytoskeletal tracks in axons, and it is well established that microtubule stabilization is a landmark of early axonal development that is sufficient to induce axon formation in vivo. Besides, microtubule stabilization alone can even lead to the transformation of mature dendrites into axons in

**eLife digest** Different regions of a neuron have distinct structures and roles. For example, each neuron has a cable-like structure called the axon that extends out of the body of the cell and carries electrical signals away from the cell body. To pass these messages on to neighboring cells, branches on the axon form connections called synapses with other neurons.

The axon lacks most of the cellular machinery needed to make proteins and other molecules that the cell needs to work correctly. Therefore, neurons must transport these materials from the cell body—where they are produced—down to the end of the axon. Specialized proteins called molecular motors carry this cargo down the axon along 'tracks' composed of filaments called microtubules. Auer, Xiao et al. have now used genetic techniques to disrupt the gene that encodes an important molecular motor, called Kif5A, in developing zebrafish larvae. The effects of this manipulation on the development of the zebrafish's visual system were then examined.

When zebrafish are a few days old, neurons in the retina—the structure at the back of the eye that responds to light—extend axons into a region of the brain called the tectum. The formation of synapses between cells in the retina and the tectum provides a pathway that enables information to travel from the eye to the brain. Auer, Xiao et al. found that in larvae that lack Kif5A, axons from the retina enter the brain about a day later than they do in normal larvae. However, when these mutant axons arrive, they produce large numbers of branches, each with the potential to form multiple synapses with cells in the tectum. However, none of the resulting synapses appear to respond to visual stimuli, which is consistent with the fact that Kif5A mutant larvae are blind.

Experiments to identify what triggers the excessive branching of retinal axons revealed that the mutant fish had elevated levels of a growth-promoting protein called neurotrophin-3 in cells in the tectum. This increased production of neurotrophin-3 was also observed when neuronal activity was blocked, for example by toxins. The lack of neuronal activity in retinal axons therefore seems to increase the production of neurotrophin-3, which in turn stimulates axonal branching. Future experiments could investigate the molecular signal that drives this increased production of neurotrophin-3, and how this is regulated during normal neuronal development.

differentiated neurons (*Gomis-Ruth et al., 2008*; *Witte and Bradke, 2008*; *Witte et al., 2008*). Along the axonal microtubule cytoskeleton molecular motors of the kinesin and dynein superfamily act as main transport molecules (*Hirokawa, 1998*; *Karki and Holzbaur, 1999*; *Vale, 2003*). Although it is well established that anterograde molecular motors are essential for synapse generation and function (*Okada et al., 1995*), little is known about their exact role in neural circuit establishment in vivo.

Of the kinesin superfamily, which comprises 45 members in mammals (*Miki et al., 2001*) and many more in zebrafish, the kinesin I subclass plays an especially prominent role in neuronal function. The kinesin motors mediate the plus end directed transport of cargo proteins along microtubules and are composed of two identical heavy chains and two identical light chains (*Hirokawa et al., 2010*). In the mammalian genome, three kinesin I heavy chain genes are present: *Kif5A*, *Kif5B*, and *Kif5C*. While Kif5B is ubiquitously expressed, Kif5A and Kif5C are neuron-specific (*Xia et al., 1998*). Their cargoes include voltage-gated potassium channels, AMPA receptor GluR2, GABAA receptors, sodium channels, neurofilaments, and mitochondria (*Rivera et al., 2007*; *Uchida et al., 2009*; *Twelvetrees et al., 2010*; *Karle et al., 2012*; *Su et al., 2013*; *Barry et al., 2014*). In humans, Kif5A mutations have been implicated in a heterogenous group of neurodegenerative disorders, including a form of Hereditary Spastic Paraplegia characterized by slowly progressive lower limb paralysis (*Goldstein, 2001*) and Charcot Marie Tooth Type 2, a peripheral axonal neuropathy (*Crimella et al., 2012*).

In this study, we have taken a combined genetic, molecular biological, and in vivo imaging approach in developing zebrafish larvae to investigate the role of anterograde intracellular transport in the development of connections between retina and tectum. By TALEN-mediated gene disruption, we generated a *kif5aa* loss-of-function allele and could show that *kif5aa* mutant fish display a de-synchronisation of retinal axon and tectal growth. A delay in tectal innervation by mutant retinal ganglion cell (RGC) axons is followed by a period of exuberant branching, resulting in enlarged axonal arbors. GCaMP imaging revealed that *kif5aa* mutant RGCs do not transmit signals from the retina to their postsynaptic partner cells. Utilizing two additional zebrafish mutant lines with defects in RGC

formation or function, *lakritz* (*Kay et al., 2001*) and *blumenkohl* (*Smear et al., 2007*), or specific silencing of RGCs by expression of botulinum toxin light chain B (BoTxLCB), we show that in all four cases the reduction of presynaptic activity leads to increased expression of neurotrophic factor 3 (Ntf3). The overabundance of this neurotrophin causes excessive branching by RGC axons. Thus, a defect in Kif5aa-mediated axonal transport has unmasked a homeostatic mechanism that adjusts axon arbor growth to levels of synaptic activity and depends on Ntf3 signaling.

## Results

### Generation of Kif5aa loss-of-function alleles by TALEN mediated gene targeting

To investigate the role of axonal transport in visual-system development, we generated a series of insertion and deletion (indel) mutations in the open reading frame (ORF) of the zebrafish anterograde motor protein Kif5aa by targeted gene disruption using transcription activator like nucleases (TALENs). Subsequently, we isolated two alleles with a 10 base pair (bp) and 13 bp deletion, respectively (*Figure 1A*), both resulting in a frameshift in the ORF from amino acid (aa) 122 onwards. This frameshift leads to a premature stop codon at position 162 of 1033aa (*kif5aa*[*162]) of the wild-type full-length protein (*Figure 1A*). In situ hybridization showed a strong expression of *kif5aa* in RGCs. In contrast, *kif5aa* mutant mRNA is dramatically down regulated in embryos starting from 24 hours post fertilization (hpf) onwards, probably by nonsense-mediated decay (*Figure 1B*). By quantitative reverse transcription PCR (qRT-PCR) (*Figure 1C*), we could see a 47% decrease of *kif5aa* transcript levels in mutant embryos compared to controls. Both alleles were not complementing each other, and the mutant phenotype co-segregated with the TALEN-induced mutation over three consecutive generations. This indicates that the genomic targeting was specific and argues for the absence of off-target effects of the TALEN pair used.

### Kif5aa mutant embryos lack visual responses and die at a larval stage

In a clutch of 5 day post-fertilization (dpf) zebrafish larvae, derived from a cross of two heterozygous carriers of the *kif5aa*[*162] allele, 25% of the larvae exhibited a dark coloration in comparison to their wild-type siblings (*Figure 1D*), indicating that the mutation is recessive, completely penetrant and results in a failure to adapt to a light background by melanosome re-distribution. This phenotype is frequently observed in visually defective mutants, specifically in those with RGC impairments (*Neuhauss et al., 1999*; *Muto et al., 2005*). For example, the *lakritz* (*Kay et al., 2001*) and the *blumenkohl* (*Smear et al., 2007*) mutations affect both vision and are darkly pigmented (*Figure 1—figure supplement 1A*). The former represents a mutation in the basic helix-loop-helix transcription factor *atonal homolog 7 (atoh7)*, which is required for RGC fate specification (*Kay et al., 2001*). Loss of Atoh7 in zebrafish leads to a complete absence of RGCs in the retina and consequently no functional connections between the retina and other brain areas are established. Nevertheless, as Atoh7 is solely expressed in the retina, no further developmental defects are described and *lakritz* mutant fish develop normally apart from their complete blindness (*Kay et al., 2001*). *Blumenkohl* mutants fail to produce a functional vesicular glutamate transporter, *vglut2a*, the main vesicular glutamate transporter expressed in zebrafish RGCs. The lack of functional Vglut2a leads to reduced synaptic transmission between the retina and the optic tectum. Furthermore, it was described that RGC axons consequently develop increased axonal arbors and show aberrant branching upon innervation of the optic tectum (*Figure 1—figure supplement 1A,C*).

To test visual system function in *kif5aa* mutant larvae, we employed the optokinetic response (OKR) to a moving grating as a sensitive and quantifiable indicator of visual functions in zebrafish (*Brockerhoff et al., 1995*). We found that the OKR was absent in *kif5aa* mutants (n = 6), while it was present in all wild-type fish examined (n = 6) (*Figure 1—figure supplement 1D*). This suggests that the disruption of *kif5aa* causes blindness. Unlike *lakritz* and *blumenkohl*, which are viable, *kif5aa* mutants fail to inflate their swim bladder (*Figure 1E*) and die around 10 days post fertilization (dpf).

As a dynamic interaction between actin- and tubulin-based motility controls melanosome transport within melanocytes (*Evans et al., 2014*), we wanted to rule out the possibility that the dark pigmentation is caused by a melanophore-autonomous defect. Kif5aa mutant embryos were treated with norepinephrin resulting in aggregation of melanosomes and consecutive re-expansion upon washing out of the drug ([*Wagle et al., 2011*], *Figure 1—figure supplement 1B*). Similar results were

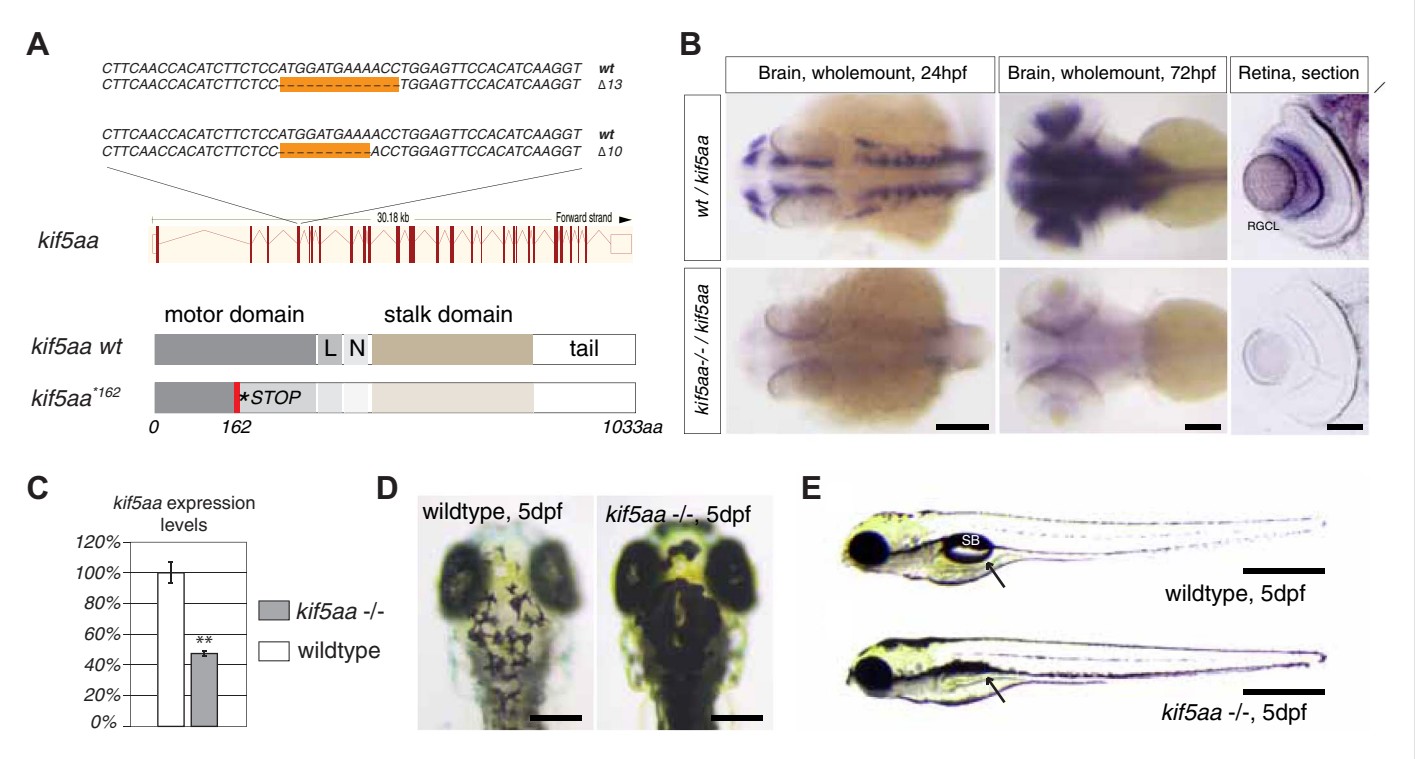

**Figure 1**. Generation of loss-of-function alleles of the anterograde motor protein Kif5aa. (**A**) Employing TALENs targeting exon4 of the *kif5aa* open reading frame, we generated two loss-of-function alleles with a 10 bp and 13 bp deletion, respectively. These result in a frameshift at amino acid 122 and a premature stop codon after 162 of 1033aas within the motor domain of Kif5aa. L = linker region, N = neck region. (**B**) *In situ* hybridization shows a substantial downregulation of *kif5aa* mRNA in 24 hpf and 72 hpf old embryos. Scale bars (from left to right) = 150 μm, 100 μm, 50 μm. RGCL = Retinal Ganglion Cell layer. (**C**) Quantitative reverse transcription PCR confirms that only 47% of wild-type *kif5aa* mRNA expression levels are reached in homozygote mutant embryos at 4 dpf (p < 0.01). (**D**) Kif5aa mutant embryos show expanded melanosomes within their melanocytes and appear dark compared to wild-type embryos. Scale bars = 200 μm. (**E**) They fail to inflate their swim bladder and die 10 days post fertilization. Scale bars = 400 μm. Arrow: pointing at the respective location of the swim bladder. SB = swim bladder.

The following figure supplement is available for figure 1:

**Figure supplement 1**. Melanosomes transport is not abolished in *kif5aa* mutants but they show no optokinetic response.

obtained for *lakritz* and *blumenkohl* (*Figure 1—figure supplement 1A*). This indicates that melanosome transport inside melanocytes in both antero- and retrograde direction is not affected in *kif5aa* mutant embryos.

## Patterning of the retina, formation of the optic chiasm, and retinotopic mapping appear normal in kif5aa mutant embryos

To examine if visual system defects were caused by retinal neurogenesis or patterning defects during development, we compared the expression of known marker genes between wild-type and *kif5aa* deficient embryos. In mutant embryos, we could not observe alterations neither in the onset of retinal neurogenesis (characterized by sonic hedgehog expression [*Shkumatava et al., 2004*]) nor in the later specification of RGCs or other retinal cell types (*Figure 2—figure supplement 1*). Choroid fissure and optic stalk formation, as well as rostral-caudal patterning, were normal as revealed by the expression of *pax2.1* and *tag-1*, respectively. Further, cell type specific marker analysis revealed that all major retinal cell types were present and the layering of the retina was not affected in *kif5aa* mutant retinae (*Figure 2—figure supplement 1*).

As retinal morphology and cellular composition were not altered by the loss of *kifaa* function, we decided to analyze the outgrowth of RGC axons from the retina and their retinotopic mapping onto

the optic tectum. To this purpose, we introduced the transgenic line *Tg(pou4f3:mGFP)* (*Xiao et al., 2005*), which labels a subpopulation of RGCs, into the *kif5aa*[*162] mutant background and imaged optic nerve outgrowth and optic chiasm formation at 48 hpf. No misrouting of RGCs to the ipsilateral side could be observed, and the optic chiasm was correctly established at the right developmental time (*Figure 2A*). To confirm that pathfinding was unaffected, we used Zn5 antibody staining to bulk-label outgrowing RGC axons (*Fashena and Westerfield, 1999*) (*Figure 2A*). Axon tracing at later stages of development, following injection of the two lipophilic dyes DiO and DiI (*Baier et al., 1996*) into opposite quadrants of the contralateral eye, revealed that the retinotectal projection was correctly patterned in mutant embryos (*Figure 2B*) and no misrouting to the ipsilateral hemisphere occurred.

## Axons enter the tectum with a delay, reminiscent of the previously identified vertigo mutant

To gain deeper insights into the phenotype of single RGCs, we made use of the *Tg(BGUG)* transgenic line (*Xiao and Baier, 2007*). The BGUG (*Brn3C* [also known as *pou4f3*]:*Gal4*; *UAS:mGFP*) transgene marks RGCs projecting to the SO (stratum opticum) and SFGS (stratum fibrosum et griseum superficiale) layers of the optic tectum. Probably due to position-effect variegation of the transgene, a stochastic subset of one to ten RGCs per retina is labeled with membrane-bound GFP, which allows the imaging of RGC trajectories in the living or fixed fish brain. By in vivo imaging, we thus followed the growth behavior of single RGC axons over consecutive days. To guarantee comparability between RGC axons, we analyzed only RGC axons growing into the SFGC layer of the optic tectum at a central

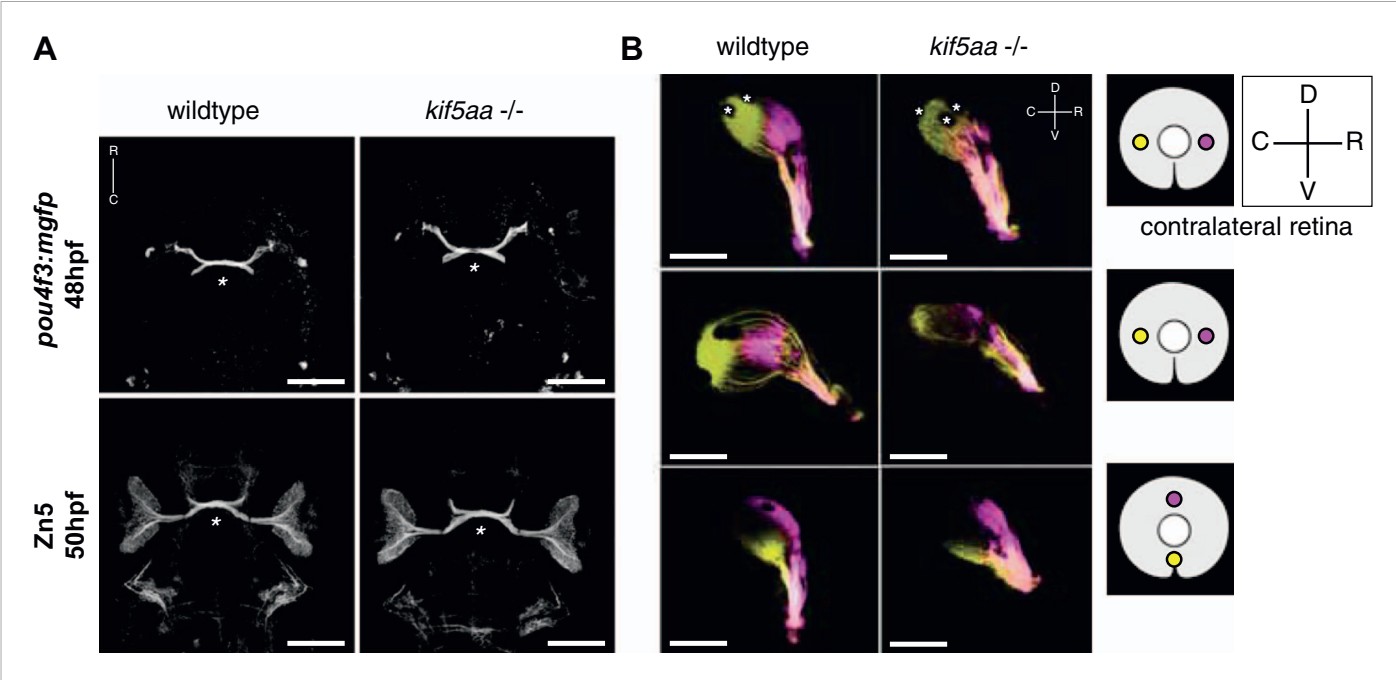

**Figure 2**. Outgrowth of the optic nerve and retinotopic mapping is normal in *kif5aa* mutants. (**A**) Confocal imaging of the *Tg(pou4f3:mGFP)* transgene, labeling a subpopulation of Retinal Ganglion Cells (RGCs) with membrane bound GFP, at 48 hpf reveals that outgrowth of the optic nerve formed by RGC axons from the retina is not affected by the *kif5aa* mutation. Immunostaining against the Zn5 antigen (DM- GRASP/neurolin present within the visual system only on RGCs [*Laessing and Stuermer, 1996*; *Fashena and Westerfield, 1999*]) confirms that optic chiasm formation is normal (marked with an asterisk). No pathfinding errors occur at this level of axonal growth. Scale bars = 200 µm. Embryos facing upwards. R = rostral, C = caudal. (**B**) Injections of the lipophilic dyes DiI and DiO in different quadrants of the contralateral retina (depicted in the right panel) show that retinotopic mapping to the optic tectum is performed in the correct manner. Asterisks = pigment cells in the skin. D = dorsal, V = ventral, R = rostral, C = caudal. No misrouting of RGC axons to the ipsilateral tectum was observed (data not shown). Scale bars = 150 µm.

The following figure supplement is available for figure 2:

**Figure supplement 1**. Patterning of the mutant retina and neurogenesis is not affected in mutants.

position on the rostral-caudal axis. While wild-type axons reach the tectal neuropil at 72 hpf and start forming a complex axonal arbor, axons of *kif5aa* mutant RGCs showed a delayed ingrowth into the target tissue (*Figure 3A*). This phenotype was confirmed by analyzing a larger RGC population, which was labeled by DiI injections into quadrants of the contralateral retina (*Figure 3A*).

A strikingly similar phenotype was reported in a previously published forward genetic screen for defects in the visual system in the *vertigo*[s1614] mutant without further in-depth characterization of this mutant line nor identification of the causal gene (*Xiao et al., 2005*). Our genetic linkage analysis confirmed that the ethylnitrosourea-induced mutation of the *vertigo*[s1614] allele is located at linkage group 9 of the zebrafish genome (*Figure 3—figure supplement 1*). Utilizing the newly generated *kif5aa*[*162] allele we could confirm by complementation crosses that *vertigo*[s1614] represents a loss-of-function allele of *kif5aa*. Although sequencing of the genome has not identified a telltale mutation of *kif5aa* coding sequence or its flanking regulatory regions in *vertigo* mutants, judging by its penetrance and expressivity, we expect the *kif5aa*[s1614] mutation to be a strong hypomorph or null allele. Furthermore, our analysis did not reveal any phenotypic difference between the different alleles and for our subsequent analysis we used *kif5aa*[*162].

## Axons of kif5aa mutants exhibit increased filopodial extensions and branch excessively

Wild-type axons continually expand their arbors during lifelong growth of the tectum. After 5 dpf, however, branching activity noticeably subsides, and axons maintain their complexity over the following days (*Meyer and Smith, 2006*) (*Figure 3B*). This phase of relative stability coincides with the consolidation of synaptic connections with tectal dendrites in the neuropil region (*Nevin et al., 2010*). In *kif5aa* mutant axons, the observed delay of ingrowth into the tectum is followed by a period of highly active growth of the axonal arbor after 5 dpf, at a stage when wild-type axons are comparatively stable. We investigated the underlying branching dynamics by multi-day single axon imaging. Between 5 and 7 dpf, mutant axons added branches at about double the wild-type rate (*Figure 3B,C*). Mutant axons also showed markedly increased numbers of active filopodia either being retracted or newly formed at any time point analyzed, as observed in 10 min timelapse movies (*Figure 3B,D*). Together, these findings suggest that, perhaps counterintuitively, absence of the motor protein Kif5aa stimulates growth of the axon arbor and maintains high filopodia activity.

## Calcium imaging reveals the disruption of presynaptic activity in kif5aa mutant RGCs

The loss of optokinetic response of *kif5aa* mutants raises the question on which level of the visual pathway the visual information processing is perturbed. To understand this better, we used genetically encoded $Ca^{2+}$ sensors that were differentially expressed in two different neuronal types of the visual system, namely in RGCs and tectal periventricular neurons (PVNs) (*Del Bene et al., 2010*; *Nikolaou et al., 2012*; *Hunter et al., 2013*). First, we probed the overall activity of the larva's visual system from 5 to 7 dpf in response to defined visual stimuli by using the *Tg(HuC:GCaMP5G)* transgenic line, in which among other neuron types, all RGCs and PVNs are labeled (*Ahrens et al., 2013*) (*Figure 4B*, left). We did observe clear stimulus-evoked $Ca^{2+}$ transients in the wild-type PVN layer and the tectal neuropil (the later contains both PVNs dendritic and RGC axonal arbors) in response to a bar (*Nikolaou et al., 2012*) moving in a caudal-to-rostral direction (*Figure 4C*, *Video 1*) across the larva's visual field. This response was almost completely absent in *kif5aa* mutants at 5 dpf (*Figure 4C*, *Figure 4—figure supplement 1*, *Video 2*). In addition, also no response was detected in older larvae (7 dpf), arguing against a delayed onset of activity in mutants, as could have been speculated based on the observed developmental delay of RGC ingrowth (*Figure 3*). Besides bars running caudal-to-rostrally, we also tested bar stimuli running in the opposite direction, bars moving in different orientations of 45° steps across the visual field as well as looming stimuli. In neither of these, *kif5aa* mutants showed a response comparable to their wild-type siblings at any developmental stage between 5 dpf and 7 dpf (data not shown). We next investigated whether this loss of $Ca^{2+}$ responses in the tectum was due to a presynaptic defect in RGC axons. For this, the same stimulation paradigms were employed in compound transgenic fish carrying *Tg(Isl2b:Gal4)* and *Tg(UAS:GCAMP3)*, which express the $Ca^{2+}$ sensor GCaMP3 in all or nearly all RGCs (*Ben Fredj et al., 2010*; *Warp et al., 2012*). Mutant RGC axons in the tectal neuropil remained unresponsive to

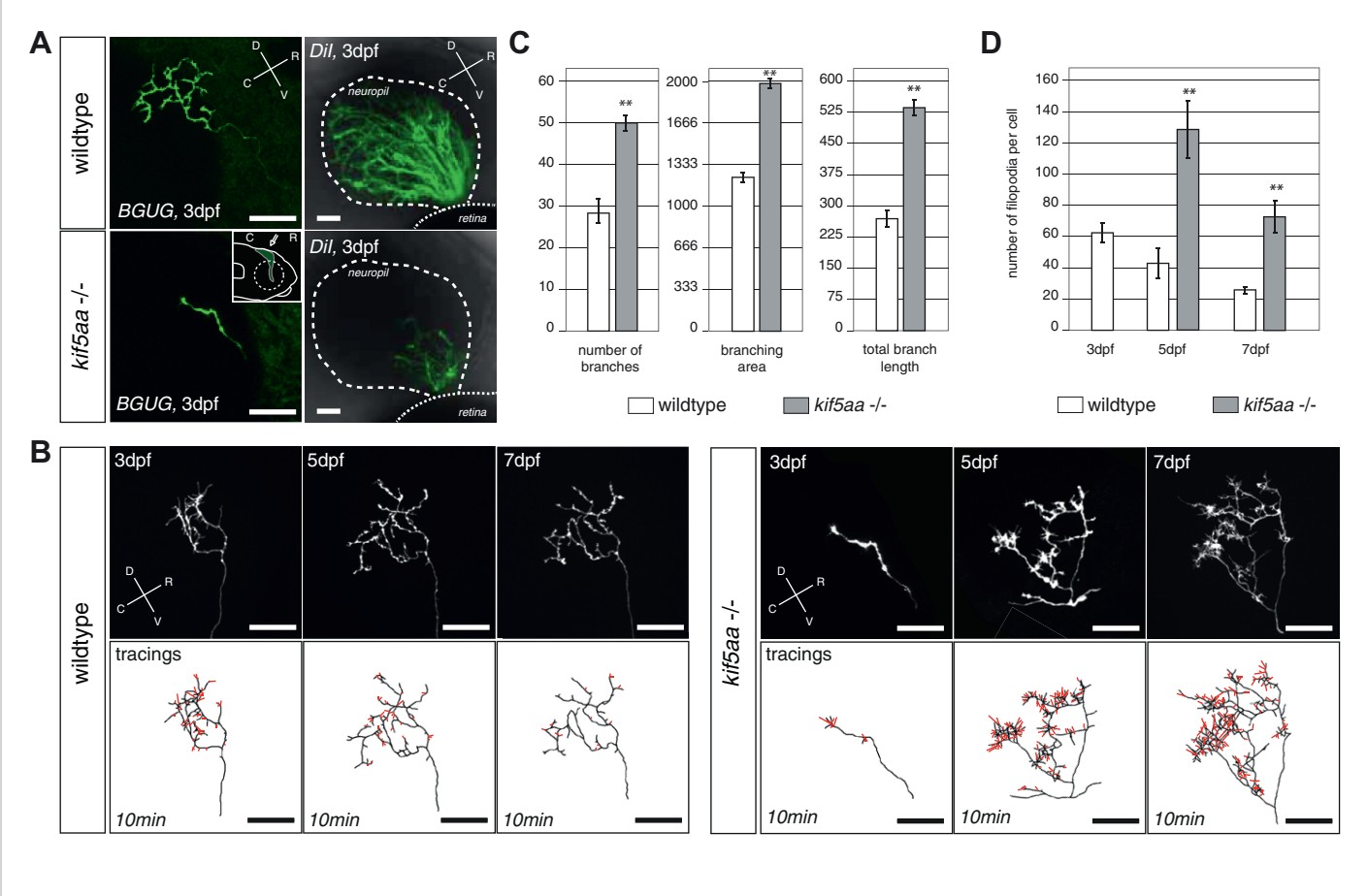

**Figure 3**. RGC axons in *kif5aa* mutants show a delayed ingrowth into the optic tectum and grow larger arbors at later stages. (**A**) Single membrane-GFP expressing RGC axons from the *Tg(BGUG)* transgene (left panel) and DiI injections into the contralateral retina of 3 dpf old wild-type and *kif5aa* mutant embryos (right panel) illustrate the delay of tectal innervation in mutants. Scale bars = 20 μm. The schematic in the lower left panel illustrates the perspective chosen for image acquisition (indicated by an arrow). D = dorsal, V = ventral, R = rostral, C = caudal. (**B**) Upper panel, left: Axonal arbor of a single wild-type RGC at 3, 5, and 7 dpf. Lower panel, left: Tracings of an axonal arbor at time point zero. In red: Overlay of filopodia formed and retracted within 10 min (1 frame/2 min). Scale bars = 20 μm. Upper panel, right: Axonal arbor of a single *kif5aa* mutant RGC axonal arbor at 3, 5, and 7 dpf. Lower panel, right: Tracing of an axonal arbor at time point zero. In red: Overlay of filopodia formed and retracted within 10 min (1 frame/2 min). Scale bars = 20 μm. (**C**) Kif5aa mutant RGC axons grow significantly more branches, cover a larger area of the optic tectum with their arbors (in μm²) and grow longer arbors (in μm) than wild-type cells at 7 dpf (n = 10, p < 0.01). Scale bars = 20 μm. For quantification, only branches stable within 10 min of image acquisition were selected. (**D**) Quantification of filopodia numbers formed and retracted within 10 min per cell at 3, 5, and 7 dpf. While wild-type RGC arbors form most filopodia at 3 dpf and reduce this rate constantly until 7 dpf, *kif5aa* mutant RGC axons grow almost three times more filopodia at 5 dpf (n = 4, p < 0.01). At 7 dpf, the rate is still more as double as high as for their wild-type counterparts (n = 4, p < 0.01).

The following figure supplement is available for figure 3:

**Figure supplement 1**. Mapping of the *vertigo*[s1614] locus by genetic linkage analysis.

visual stimulation, whereas wild-type axons were robustly activated (*Figure 4D*). These data provide an explanation for the behavioral blindness of *kif5aa* mutant larvae and indicate that the Kif5aa motor is required for proper synaptic transmission from RGC axon terminals to tectal dendrites.

## Kif5aa mutant RGCs form presynaptic sites but are depleted of mitochondria

As loss of synaptic activity in RGCs might be caused by a failure in synapse formation, we monitored synapse distribution and transport of synaptic vesicles in RGC axons in vivo. For this experiment, we

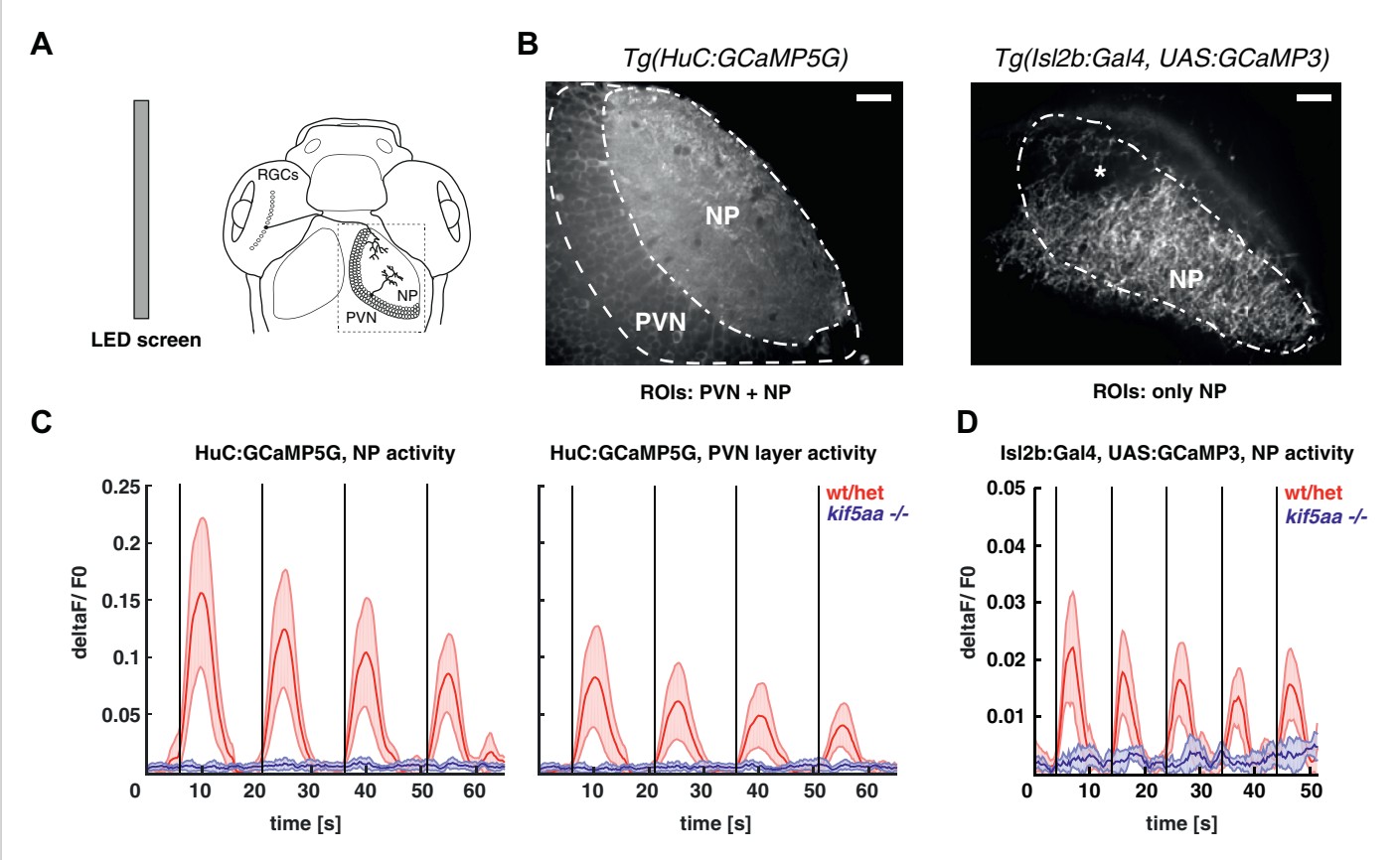

**Figure 4**. Kif5aa mutant larvae show no activity in RGCs and no synaptic transmission to tectal cells. (**A**) 5–7 dpf larvae were visually stimulated by bars on an LED screen running in caudal-to-rostral direction across the larva's visual field. Wild-type and *kif5aa* mutant larvae expressing genetically encoded calcium indicators (GCaMPs) in different subsets of neurons of the visual system were confocally imaged in the tectum contralaterally to the stimulated eye (dashed box inset). RGCs = Retinal Ganglion Cells, PVNs = periventricular neurons, NP = neuropil. (**B**) The activity of visual system neurons in response to visual stimuli is shown as normalized GCaMP fluorescence intensity changes (deltaF/F0) over time. GCaMP intensity was averaged over manually determined regions of interest (ROIs) that corresponded to well-distinguishable anatomical regions in the larval tectum, the neuropil (NP) and the periventricular cell bodies area (PVNs). In *Tg(HuC:GCaMP5)* fish (left), GCaMP5 is expressed pan-neuronaly, that is, in both neuropil and PVNs, whereas in *Tg(Isl2b:Gal4)* × *Tg(UAS:GCaMP3)* fish (right), it is expressed in all RGCs and their processes. Scale bars = 20 µm. Asterisk = pigment cell in the skin. (**C**) Averaged deltaF/F0 ratio over time in response to a moving bar visual stimulation (black vertical line denotes the time point of the stimulus onset) in fish with pan-neuronal GCaMP5G expression. Four sequential rounds of stimulus presentation and the time-courses of Ca²⁺-transients in the neuropil ROIs (left panel) and periventricular cell bodies ROIs (right panel) of wild-type/heterozygous (red curve) and *kif5aa* mutant larvae (blue curve) (n = 7 each) are shown. Activity in the visual system of *kif5aa* mutants was almost absent between 5 and 7 dpf compared to wild-type larvae. Light red and blue zones indicate the 95% confidence intervals around the averaged deltaF/F0 curves for each region of interest (NP and PVN), respectively. (**D**) Averaged deltaF/F0 ratio over time in response to a moving bar visual stimulation (black vertical line denotes stimulus onset) in fish with GCaMP3 expression in RGC axons. Five sequential rounds of stimulus presentation and time-courses of Ca²⁺-transients in the neuropil of wild-type/heterozygous (red curve) and *kif5aa* mutant larvae (blue) (n = 9 each). RGC arbor activity in *kif5aa* mutants was strongly diminished. Light red and blue zones indicate the 95% confidence intervals around the averaged deltaF/F0 curves for the neuropil region of interest (NP).

The following figure supplement is available for figure 4:

**Figure supplement 1**. Regression-based analysis of wild-type/heterozygous vs *kif5aa*–/– *Tg(HuC:GCaMP5G)* mutants to a visual stimulus.

made use of a Synaptophysin-GFP (Syp-GFP) fusion construct, a marker for stable presynaptic sites as well as for motile Synaptophysin-containing clusters (*Meyer and Smith, 2006*). When co-expressed with a membrane-localized red fluorescent protein (RFPCaax) in single RGCs in wild-type or *kif5aa* mutant embryos (*Figure 5A*), we observed no difference in distribution of stable presynaptic clusters at 5 and 7 dpf by in vivo imaging (*Figure 5B*). This indicates that the formation of stable presynaptic clusters is not affected by the loss of Kif5aa. Quantification of Synaptophysin-containing vesicle

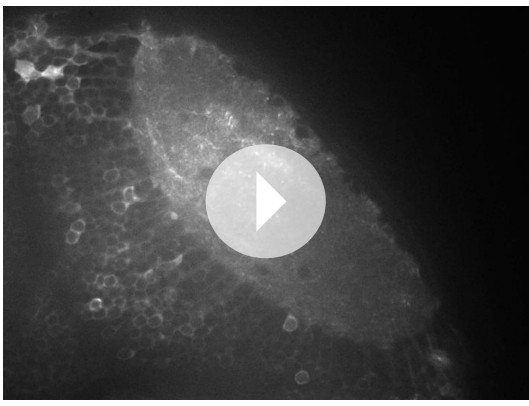

**Video 1.** In vivo timelapse imaging of the optic tectum in a *Tg(HuC:GCaMP5G)* transgenic 5 dpf wild-type fish. A wild-type larva was stimulated with a visual bar running from caudal to rostral through the visual field of the contralateral eye to the imaged optic tectum. Stimulus onset is indicated by a white circle in the top right corner of the image sequence. The movie is accelerated to a framerate of 40 frames/s.

movements in axonal segments (*Figure 5—figure supplement 1*, *Video 3*) did not show a difference in either anterograde or retrograde transport upon loss of *kif5aa* at 4, 5, and 7 dpf. This is consistent with previous studies showing that Synaptophysin containing vesicles are not transported by Kinesin I (*Karle et al., 2012*) and furthermore argues against a general loss of vesicle movement in *kif5aa* mutant RGCs. To distinguish between different sized clusters, we divided vesicles into small (<0.4 µm) and middle-sized to large clusters (>0.4 µm) as defined previously for RGC axons in zebrafish (*Meyer and Smith, 2006*). Using these different categories, we detected a higher fraction of small, motile clusters in *kif5aa* mutant cells at 4 and 5 dpf, but not at 7 dpf, most probably reflecting their highly active growth behavior at that stage of development (*Figure 5—figure supplement 1C*).

In vivo imaging of labeled mitochondria (mitoGFP) and membranes (RFPCaax) in single RGCs in contrast showed a reduction of mitochondria within the axon of mutant RGCs compared to wild-type cells (*Figure 5C*). This is consistent with previous reports showing that mitochondria are transported by Kif5a in other experimental systems (*Macaskill et al., 2009*; *Karle et al., 2012*; *Chen and Sheng, 2013*). We confirmed this result by quantification of the area covered by mitochondria per neuropil area in electron micrographs of transverse section of the tectal neuropil (*Figure 5D*). At 6 dpf, mutants show a significant depletion of mitochondria from their branched axons (*Figure 5E*). To test if this mitochondria depletion from the distal axon of RGCs is caused by transport defects of this known cargo of Kif5a, we quantified the

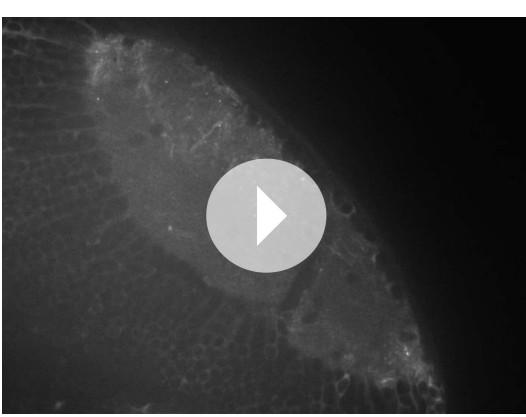

**Video 2.** In vivo timelapse imaging of the optic tectum in *Tg(HuC:GCaMP5G)* transgenic 5 dpf *kif5aa* mutant embryos. A *kif5aa* mutant larva was stimulated with a visual bar running from caudal to rostral through the visual field of the contralateral eye to the imaged optic tectum. Stimulus onset is indicated by a white circle in the top right corner of the image sequence. The movie is accelerated to a framerate of 40 frames/s.

transport dynamics of mitochondria within RGC axonal segments (*Figure 5—figure supplement 2*, *Video 4*). We did not detect a difference in overall amount of mobile vs stable mitochondria. In mutant cells, though, mitochondria were transported significantly more often in a retrograde direction than in an anterograde direction (*Figure 5—figure supplement 2D*). This bias explains the depletion of mitochondria from the tips of *kif5aa* mutant RGCs. Mitochondria are preferentially localized at active synapses (*Obashi and Okabe, 2013*) and are found in close proximity to stable Synaptophysin-containing clusters in RGC arbors (*Figure 5—figure supplement 2A*). By co-labeling mitochondria and presynaptic clusters in the same cells in vivo we showed that approximately 40% stable presynaptic sites are associated with mitochondria in wild-type and *blu–/–* RGC axons (*Figure 5—figure supplement 2B*). We extended and confirmed these data observing mitochondria distribution and presynaptic densities in trigeminal ganglion cell axons (TGCs) (*Figure 5—figure supplement 2A,B*). In contrast to wild-type cells, in TGCs in *kif5aa–/–* we observed a marked

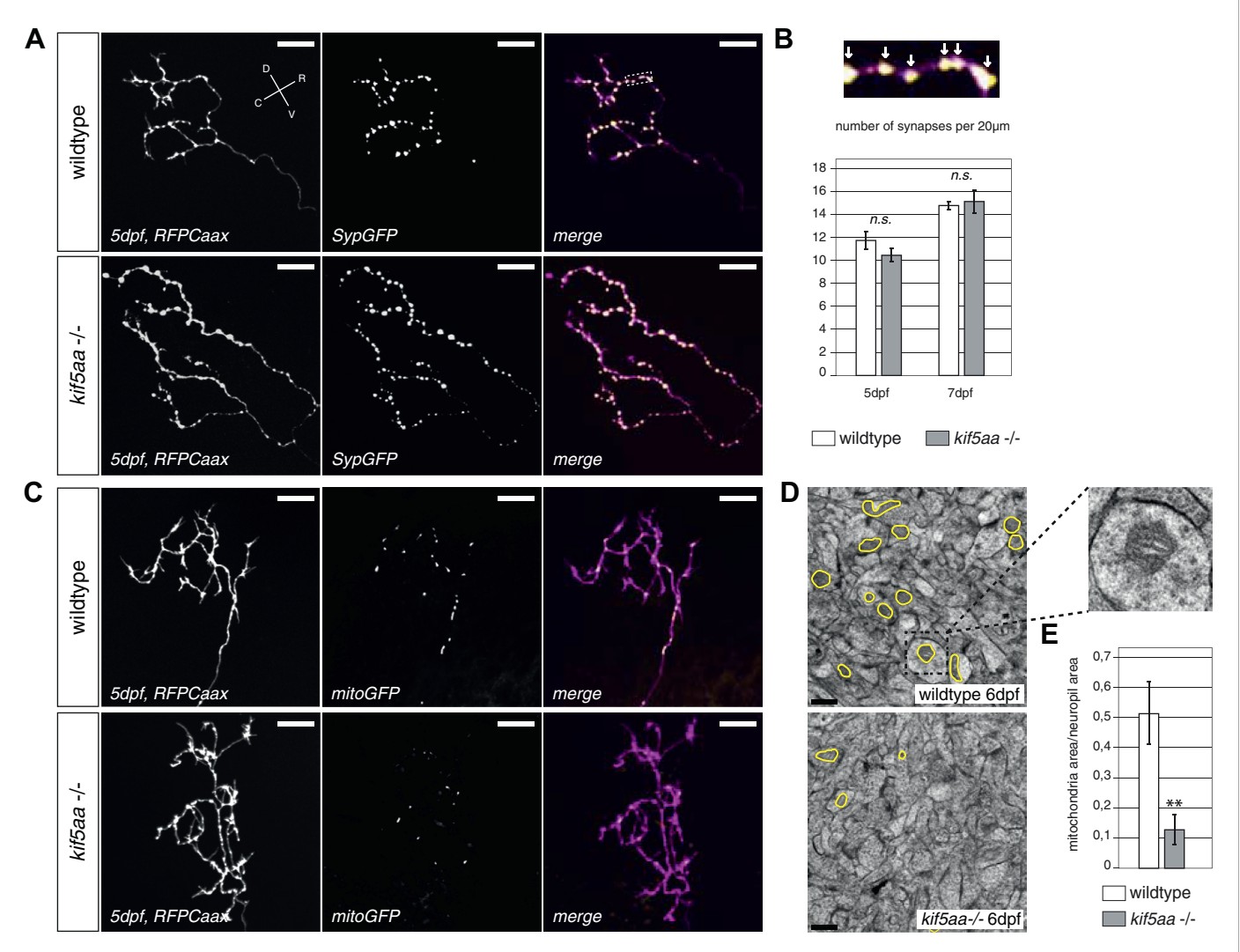

**Figure 5**. Kif5aa mutant RGC arbors show the same density of presynaptic sites but are depleted of mitochondria. (**A**) In vivo imaging shows the distribution of presynaptic sites marked by Synaptophysin-GFP (SypGFP) in single *kif5aa* mutant and wild-type RGC arbors expressing membrane localized RFP (RFPCaax). Upper panel: wild-type cell arbor, lower panel: *kif5aa* mutant cell arbor. Scale bars = 20 μm. D = dorsal, V = ventral, R = rostral, C = caudal. (**B**) Upper panel: Zoom in to an axonal segment indicated in the right panel of (**A**). Stable presynaptic clusters of SypGFP larger than 0.4 μm were defined as synapses (white arrows in the upper panel) and synapse density in axonal segments of wild-type and mutant cell arbors does not show a significant difference at 5 and 7 dpf (lower panel). (**C**) Distribution of mitochondria (labeled by mitoGFP) in single mutant and wild-type RGC arbors expressing membrane localized RFP (RFPCaax) in vivo. Upper panel: wild-type cell arbor, lower panel: *kif5aa* mutant cell arbor. Mutants RGC cell arbors show substantially less mitochondria. Scale bars = 20 μm. (**D**) Transmission electron micrograph of a transvers section of the neuropil containing RGC axonal arbors. Upper panel: wild-type neuropil, lower panel: *kif5aa* mutant neuropil. In yellow circles: mitochondria. Left panel: Zoom in into a single axonal segment containing a mitochondrion. The *kif5aa* mutant neuropil contains less mitochondria. Scale bar = 500 nm. (**E**) Quantification of mitochondria area per neuropil area comparing wild-type and mutant tecta at 6 dpf. Mutant cells contain significantly less mitochondria than wild-type cells (p < 0.01).

The following figure supplements are available for figure 5:

**Figure supplement 1**. Analysis of transport dynamics of synaptic vesicles in wild-type and *kif5aa* mutant cells during visual system development.

**Figure supplement 2**. Analysis of mitochondria localization and transport dynamics in wildtype and *kif5aa* mutant RGC arbors.

**Figure supplement 3**. Retinal Ganglion Cells show a normal mitochondria distribution in *blumenkohl* mutants.

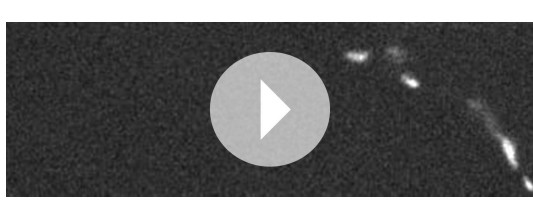

**Video 3.** In vivo timelapse imaging of Synaptophysin-GFP containing clusters in RGC axonal segments. Representative RGC axonal segment in a 5 dpf old wild-type larva. SypGFP labels synaptic clusters of different sizes that were grouped in small and middle-sized plus large vesicles. Compare *Figure 5—figure supplements 1A,B* for grouping and kymogram analysis.

decrease of presynaptic densities located in close proximity of mitochondria. This observation is consistent with a recent study in zebrafish reporting a reduced number of mitochondria in peripheral cutaneous axon arbors in *kif5aa* mutant zebrafish embryos (*Campbell et al., 2014*) without affecting the distribution of presynaptic densities.

Taken together, these experiments show that *kif5aa* mutant RGCs form presynaptic sites at the same density as wild-type cells and transport Synaptophysin-containing clusters at the same rate and direction as wild-type. A detectable reduction in the relative anterograde transport of mitochondria, however, results in a depletion of these organelles from synaptic terminals.

## Lack of synaptic input leads to an increased production of neurotrophic factor 3 (Ntf3) in the tectum

We aimed to identify the signal responsible for the observed increased growth of RGC arbors in *kif5aa* mutant tecta. As it was previously shown in the optic tectum of *X. laevis* that Brain-Derived Neurotrophic Factor (BDNF) can promote axonal arborization (*Cohen-Cory and Fraser, 1995*), we decided to measure the expression levels of this neurotrophic factor in *kif5aa* mutant embryos. In parallel, we performed quantitative reverse transcription PCR of other known members of the neurotrophic factor family, namely *neurotrophin 3* (*ntf3*), *neurotrophin 4* (*ntf4*), *neurotrophin 7* (*ntf7*), and *nerve growth factor* (*ngf*) present in the zebrafish genome (*Heinrich and Lum, 2000*). Comparing 4 dpf old homozygous *kif5aa* mutants to their siblings, we detected a significant upregulation of *ntf3* in mutants to up to 160% of its normal expression level in wild-type embryos, while the levels of *bdnf*, *ngf*, *ntf4*, and *ntf7* were not significantly altered (*Figure 6A*). By in situ hybridization with an *ntf3* antisense probe, we could furthermore see an increased staining intensity in mutant tecta compared to their siblings (*Figure 6B*).

For further validation of these results, we performed Western blotting analysis using an antibody targeting the human orthologue of Ntf3. This antibody recognizes the zebrafish Ntf3 protein. We could demonstrate cross-reactivity by overexpressing a construct carrying the zebrafish *ntf3* cDNA followed by an E2A sequence allowing multicistronic expression (*Szymczak et al., 2004*) of *ntf3* and a RFP reporter gene from the same cDNA (*Figure 6C*). This experiment showed that *kif5aa* mutant embryos produced significantly more Ntf3 protein than their siblings.

To see if Ntf3 upregulation was a common feature in mutants with defective retinotectal synaptic transmission, we investigated *lakritz* mutants, which lack all RGCs and *blumenkohl*, in which RGCs show impaired glutamate secretion into the synaptic cleft. Ntf3 protein was increased in the tecta of all three mutant lines in which presynaptic input to the tectum was abolished or highly reduced (*Figure 6D*). To directly test the causality link between the lack of presynaptic input and upregulation of Ntf3 in the optic tectum, we silenced the neuronal activity of RGCs by expressing a UAS:BoTxLCB-GFP construct in Islet2b:Gal4 zebrafish larvae. BoTx has been shown to specifically block synaptic vesicle release (*Brunger et al., 2008*) and its injection has been used successfully in zebrafish embryos to silence neuronal activity (*Nevin et al., 2008*). Similar to what previously observed in *kif5aa, lak*, and *blu* mutants, silencing of RGCs via BoTx expression leads to melanosomes expansion and failure to adapt to a light background (*Figure 6—figure supplement 1A*). Both, via qRT-PCR and Western blotting analysis we detected an upregulation of Ntf3 in these transgenic animals showing that, like in the previously described mutants, lack of RGCs presynaptic activity per se is sufficient to cause Ntf3

**Video 4.** In vivo timelapse imaging of mitochondria in RGC axonal segments. Representative RGC axonal segment in a 5 dpf old wild-type larva. Compare *Figure 5—figure supplement 2B* for kymogram analysis of mitochondria movements.

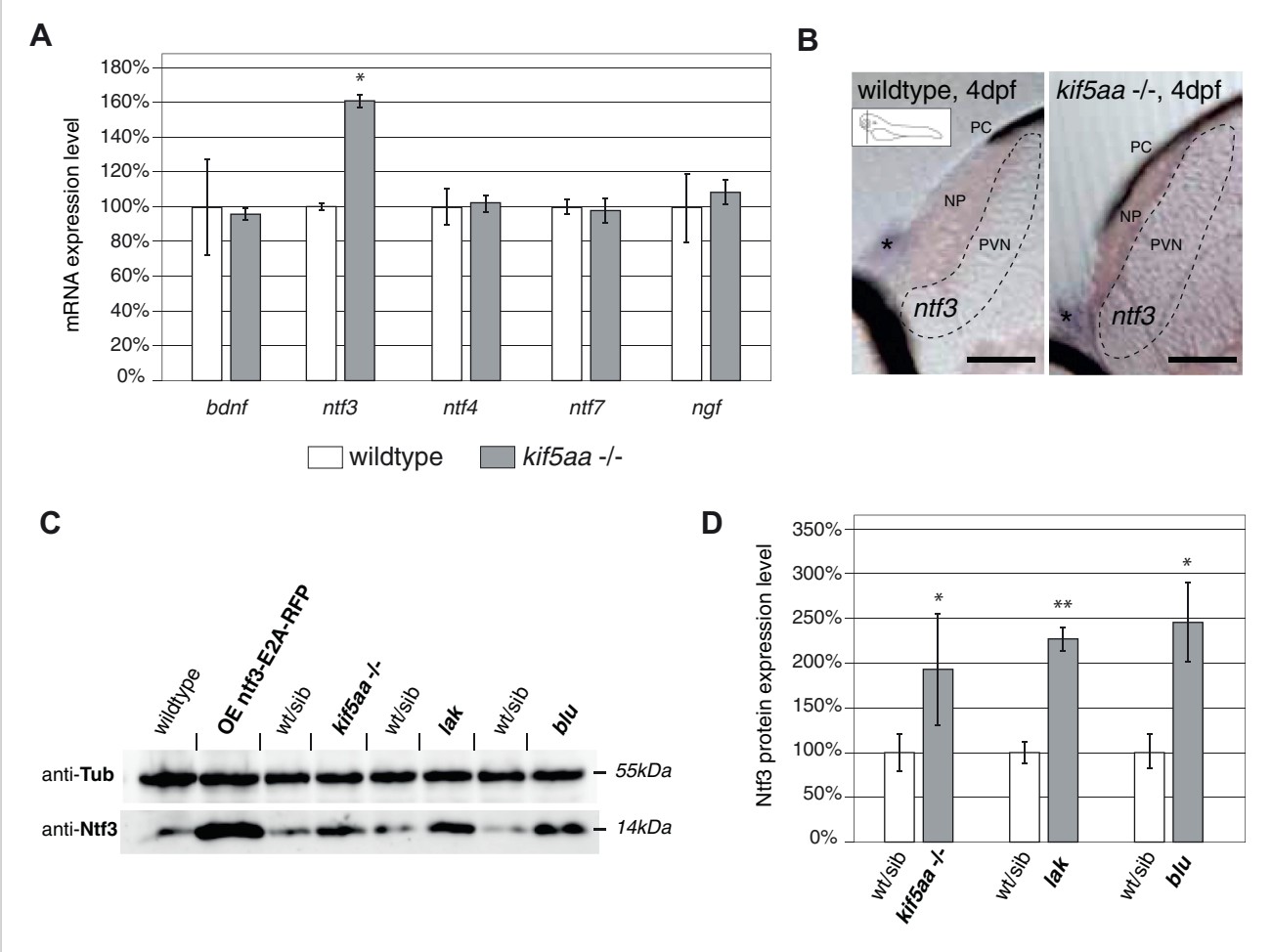

**Figure 6**. Expression of the neurotrophic factor neurotrophin 3 in visually impaired mutants. (**A**) Relative expression levels of *bdnf*, *ntf3*, *ntf4*, *ntf7*, and *ngf* in 4 dpf old wild-type and *kif5aa* mutant embryos. *ntf3* is upregulated to 160% of wild-type expression levels (p < 0.05) while all other neurotrophic factors show the same expression levels between wild-type and mutant embryos. (**B**) Transversal sections through the tectum after in situ hybridization with an *ntf3* specific antisense probe detect higher expression levels of *ntf3* in tecta of *kif5aa*. Scale bars = 25 μm. PC = pigment cell, NP = neuropil, PVN = periventricular neurons. Asterisk = strong *ntf3* signal in the otic vesicle. (**C**) Confirmation of Ntf3 overexpression by Western blotting in 4 dpf old embryos. To show the specificity of the antibody, we generated a Ntf3 overexpression construct (UAS:ntf3-E2A-RFP). The two visually impaired mutant lines *lakritz* and *blumenkohl* also show a substantial upregulation of *ntf3* expression levels. (**D**) Quantification of Ntf3 protein expression levels based on Western blotting data in wild-type and visually impaired mutant embryos. All three mutant lines show a substantial upregulation of Ntf3 protein levels.

The following figure supplement is available for figure 6:

**Figure supplement 1**. Silencing of all Retinal Ganglion Cells by BoTx expression leads to Ntf3 upregulation.

overexpression (*Figure 6—figure supplement 1B,C*). These results strongly suggest that lack of presynaptic activity and subsequent overexpression of Ntf3 in the tectum trigger the increased size of axonal branches of RGCs in *kif5aa*, *blumenkohl*, and *lakritz* mutants (see *Figure 2*, [*Smear et al., 2007*; *Gosse et al., 2008*]).

## Neurotrophic factor 3 signaling acts on RGC axonal branching

To further test this hypothesis, we designed an experiment to interfere with Ntf3 signaling in RGCs. We generated a kinase-dead, GFP-tagged, dominant-negative version of the zebrafish *ntrk3a* gene orthologous to the gene encoding the TrkC receptor in mammals (ntrk3adN-GFP) (*Parada et al., 1992*). A similar approach was previously shown to efficiently block Ntf3 function in mammalian cells (*Tsoulfas et al., 1996*; *Lin et al., 2000*). In zebrafish two TrkC paralogues, *ntrk3a* and *ntrk3b* exist,

which are both expressed in RGCs (*Figure 7A*). We decided to generate our dominant-negative construct based on the *ntrk3a* coding sequence, which shows a higher degree of conservation (based on amino acid identity and sequence similarity) to the rat TrkC receptor (*Martin et al., 1998*). Both receptors are predicted to bind to Ntf3 based on binding motif analysis (*Martin et al., 1998*). Upon overexpression of the truncated *ntrk3adN-GFP* construct in single wild-type RGCs by mosaic DNA expression, we observed a substantial reduction of axon branch length and number of branches at 5 dpf (*Figure 7B,E*), consistent with a role of Ntrk3 as a branch-promoting receptor.

In addition, *ntrk3adN-GFP* overexpression in single *kif5aa−/−* and *blu−/−* RGCs could abolish the axonal overgrowth normally observed in these mutants as measured by total branch length (*Smear et al., 2007*) and even reduce the number of branches compared to wild-type cells (*Figure 7E*).

To analyze the consequence of Ntf3 overexpression, we injected an *UAS:Ntf3-E2A-RFP* construct into the *Tg(gSA2AzGFF49A)* (*Muto et al., 2013*) transgenic line, driving the expression of the transgene in tectal neurons and glia cells from 2 dpf onwards. Thereby, we generated larvae specifically overexpressing Ntf3 in the optic tectum just before innervation by RGC axons. Analyzing the morphology of single RGCs marked by membrane-bound eGFP and growing under these conditions, we confirmed that these cells formed larger axonal arbors than wild-type cells (*Figure 7C,E*) at 5 and 7 dpf. Taken together, these results strongly support the hypothesis that Ntf3 signaling is a signal promoting RGC arbor growth and branching and that Ntf3 upregulation is responsible for axonal arbor overgrowth when RGC presynaptic activity is impaired.

We next decided to test if the observed increased formation and retraction of filopodia in the axons of *kif5aa* mutant RGCs was directly caused by Ntf3 upregulation. Therefore, we analyzed filopodia dynamics via time-lapse imaging both in *blu−/−* RGCs and in axons growing when Ntf3 was overexpressed in the tectum via our transgenic construct. In both experimental conditions we did not observe any increase in the filopodia dynamics or RGC axons (*Figure 7—figure supplement 1A–C*), excluding the possibility that Ntf3 overexpression has a direct effect on this process. In addition, we observed that the distribution of mitochondria was not significantly altered in *blu−/−* RGC axons nor was the association with stable presynaptic sites (*Figure 5—figure supplements 2B, 3*). Together these data suggest that Ntf3 upregulation does not per se affect mitochondria localization.

## Axons of wild-type RGCs transplanted into kif5aa mutant tecta show excessive branching

For the dissection of cell-autonomous vs non-cell autonomous effects of the loss of *kif5aa*, we generated mutant/wild-type chimeric embryos by blastomere transplantations at the 1000-cell stage (*Gosse et al., 2008*). Donor RGCs were derived from a *Tg(Pou3f4:Gal4) × Tg(UAS:RFP)* cross and visualized by the expression of a fluorescent membrane-targeted RFP. When cells were transplanted in low numbers from a transgenic donor into a host embryo, we could image single donor derived and fluorescently labeled RGCs, within a host environment. First, we could observe that the delayed ingrowth of *kif5aa* mutant RGC axons in the tectal neuropil is a cell-autonomous effect (*Figure 8—figure supplement 1*). When growing in a wild-type environment, *kif5aa* mutant RGC axons invade the tectal neuropil 1 day later (4 dpf) than wild-type cells. Second, in contrast to what we observed in mutant larvae, arbor size was not increased at 5 or 7 dpf in mutant RGC axons that grow into a wild-type tectum (*Figure 8A,B*). Mutant RGC arbors are significantly smaller, similar to those that overexpress *ntrk3adN-eGFP* (see *Figure 7B*). Wild-type cells growing in a *kif5aa* mutant background show the opposite behavior. No stalling at 3 dpf was detected (*Figure 8—figure supplement 1*) and, at 5 and 7 dpf, they established a significantly increased axonal arbor (*Figure 8A, B*). To further demonstrate that the axonal arbor overgrowth observed both in *kif5aa−/−* and *blu−/−* was due to a common molecular mechanism, we transplanted single *kif5aa−/−* RFP labeled RGC into a *blu* mutant host. In these conditions, *kif5aa* mutant RGC axonal arbors were significantly larger than when transplanted into a wild-type host (*Figure 8—figure supplement 2*).

Taken together, these experiments support a homeostatic mechanism by which tectum-secreted Ntf3 directly promotes the growth of innervating RGC axons. Lack of retinal synaptic activity results in upregulation of Ntf3 and, consequently, in an enlargement of RGC axonal arbors.

## Discussion

Here, we report for the first time the role of the anterograde transport motor Kif5aa in the larval development of the zebrafish visual system. The loss-of-function *kif5aa* allele that we generated

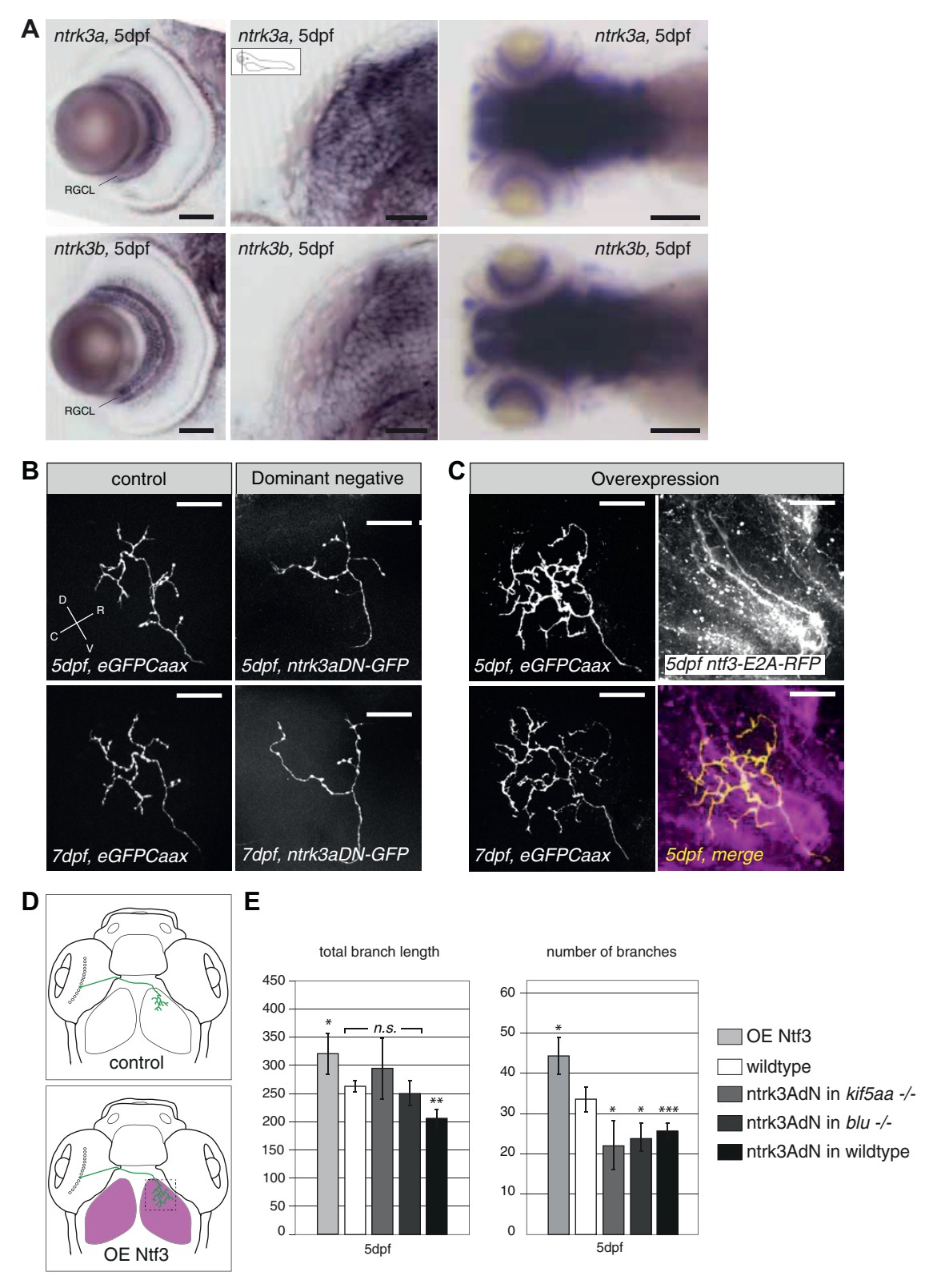

**Figure 7**. Neurotrophin 3 signaling alters axonal branch size in RGCs. (**A**) In situ hybridization with *ntrk3a* and *ntrk3b* specific antisense probes shows expression of both paralogues in broad parts of the nervous system in 5 dpf larvae. Both receptors are strongly expressed in RGCs. RGCL = Retinal Ganglion Cell Layer. Scale bars (from left to right) = 50 μm, 50 μm, 150 μm. (**B**) Representative pictures of single RGC axons at 5 (upper) and 7 dpf (lower

*Figure 7. continued on next page*

*Figure 7. Continued*

panel). Control RGC cells express a membrane bound eGFP (control; left panel). To render cells unresponsive to the TrkC pathway, single RGCs express a dominant negative, kinase dead and eGFP-tagged form of the neurotrophic factor receptor ntrk3a (ntrk3adN-GFP) (dominant negative, right panel). Consequently, RGCs grow smaller arbors with less branches. (**C**) To investigate the effect of Ntf3 on RGC axonal growth, we monitored single eGFP positive RGCs while growing into a tectum overexpressing Ntf3 (overexpression) at 5 (upper left panel) and 7 dpf (lower left panel). Overexpression of Ntf3 was driven by an UAS:ntf3-E2A-RFP construct in the *Tg(gSA2AzGFF49A)* (*Muto et al., 2013*) transgenic line in tectal glial cells and periventricular neurons from 2 dpf onwards. By employing a 2A sequence between the *ntf3* and the RFP open reading frame, both proteins were produced from the same construct. Thereby Ntf3 overexpressing cells were marked by RFP expression (right upper panel) and we analyzed the arbors of single eGFPCaax positive RGCs at 5 (upper left panel) and 7 dpf (lower left panel) growing in RFP expressing optic tecta. RGCs grow more complex arbors with more branches when invading into the Ntf3 overexpressing tectal environment compared to control RGCs (**A**, left panel). Lower left panel = merge of ntf3-E2A-RFP expressing tectal cells and an eGFPCaax expressing RGC axon. Scale bars = 20 µm. D = dorsal, V = ventral, R = rostral, C = caudal. (**D**) Schematics illustrating the approach for analysis of single RGC arbors. While in control and dominant negative expression experiments, single RGCs were labeled (upper panel), in the Ntf3 overexpression situation, single membrane bound eGFP (eGFPCaax) labeled RGCs were growing into a tectum overexpressing Ntf3 (labeled by RFP expression, shown in magenta, lower panel). (**E**) Quantification of total branch length and number of branches at 5 dpf in single RGC arbors upon overexpression of the dominant negative ntrk3adN-GFP construct in single RGCs or overexpression of Ntf3 in tectal cells. Ntrk3adN-GFP expressing wild-type cells are significantly smaller and grow fewer branches at 5 dpf (p < 0.001). In both, kif5aa and blumenkohl mutant embryos, expression of ntrk3adN-GFP in single RGCs inhibits the overgrowth of the axonal arbor that is normally observed. The branch length is not different to the length in wild-type cells. Ntf3 overexpression in the tectum leads to increased axonal branch length and increased branch number in wild-type RGCs (p < 0.05) (n = 8, 23, 7, 8, 26).

The following figure supplement is available for figure 7:

**Figure supplement 1**. Blumenkohl mutant RGC arbors and RGC arbors growing into a Ntf3 overexpressing tectum do not show increased filopodia dynamics.

disrupts the open reading frame after 122 of 1033aa within the motor domain of the protein and results in mRNA degradation likely by nonsense-mediated decay. We therefore expect that no functional Kif5aa protein is produced.

Disruption of Kif5aa created a complex and dynamically changing retinotectal phenotype. At 3 to 4 dpf, RGC axons devoid of Kif5aa grew more slowly and reached their targets in the tectum with a delay of about 24 hr. Mutant retinotectal synapses did not transmit signals to tectal cells, but were apparently silent, resulting in a complete loss of visual responses. Both the stalled growth and the synaptic transmission defects were likely a direct consequence of the absence of Kif5aa. Some of the known cargoes of mammalian Kif5a are required for normal axon outgrowth (*Karle et al., 2012*; *Chen and Sheng, 2013*; *Schwarz, 2013*; *Sheng, 2014*). In the zebrafish *kif5aa* mutant, mitochondria are significantly depleted from distal RGC axon terminals, as shown by in vivo imaging and transmission electron microscopy, suggesting that these organelles are Kif5a cargoes as in mammals. Furthermore, a recent report showed that Kif5aa has a similar role in the posterior lateral line nerve and peripheral cutaneous axonal arbors (*Campbell et al., 2014*). Mitochondrial ATP production is required for synapse assembly (*Lee and Peng, 2008*), the generation of action potentials (*Attwell and Laughlin, 2001*) and synaptic transmission (*Verstreken et al., 2005*). In addition, synaptic mitochondria maintain and regulate neurotransmission by buffering $Ca^{2+}$ (*Medler and Gleason, 2002*; *David and Barrett, 2003*). This deficit can therefore explain the impairment in transmitter release at the presynaptic terminals without, however, excluding the possibility that the impaired transport of other cargoes is also involved.

Interestingly, outgrowth of axons from the retina was not affected by loss of Kif5aa. Only after crossing the optic chiasm, at the entrance to the tectum, did RGC axons stall. Reduced axonal growth is caused by loss of kinesin-1 in other experimental systems or other axonal transport motors like Dynein/Dynactin (*Ferreira et al., 1992*; *Ahmad et al., 2006*; *Abe et al., 2008*; *Karle et al., 2012*; *Prokop, 2013*), but often the deficit is evident from the start. Early functions of kinesin I heavy chains are probably carried out by other members of this gene family in zebrafish, such as *kif5ab*, *kif5b* or *kif5c*. All isoforms homodimerize and may carry distinct sets of cargoes (*DeBoer et al., 2008*). In mammalian neurons, KIF5C likely contributes to axon specification (*Jacobson et al., 2006*). Similarly, the *Drosophila* KIF5 homolog kinesin heavy chain (KHC) drives axon initiation and transiently maintains axonal growth (*Lu et al., 2013*). The zebrafish Kif5c orthologue is therefore the prime candidate to carry out this early kinesin motor function for RGCs.

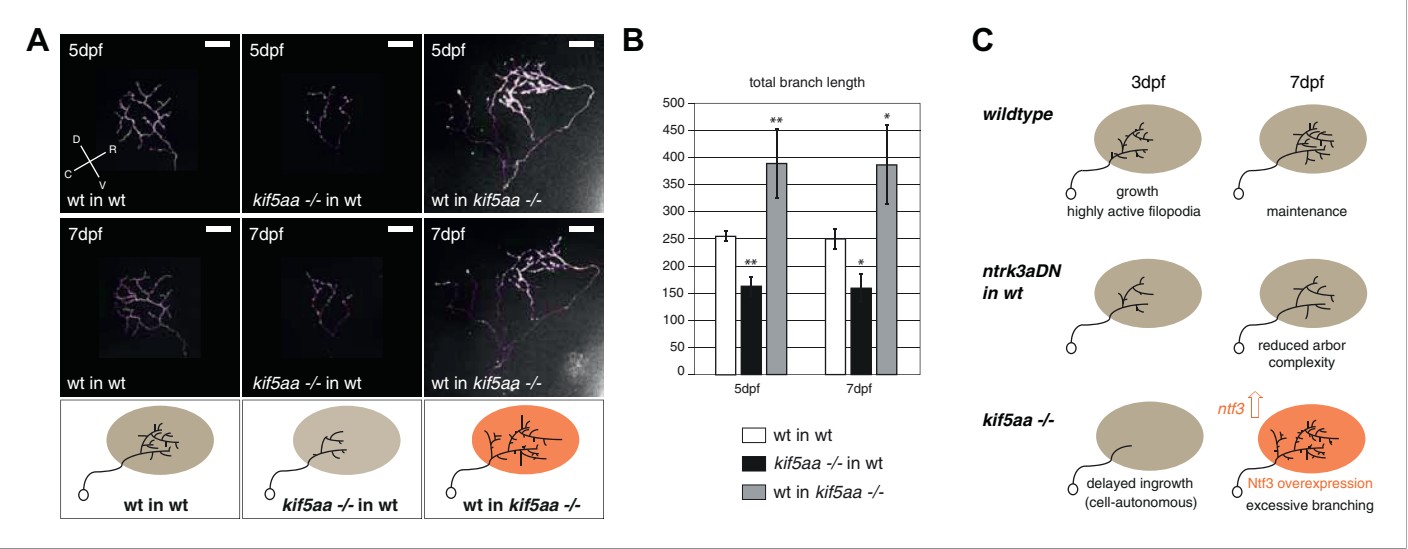

**Figure 8**. Transplantations confirm the growth promoting-effect in *kif5aa* mutant tecta. (**A**) Representative pictures of single in vivo imaged RGC axons after blastula stage transplantions from wild-type donors into a wild-type tectum (left panel), from *kif5aa* mutants into a wild-type tectum (middle panel) or from a wild-type donor into a *kif5aa* mutant tectum (right panel). The same cell was analyzed at 5 dpf (upper panel) and 7 dpf (middle panel). Scale bars = 20 μm. Schematics of RGC arbor complexity and size in the lower panel. In orange: Ntf3 overexpressing *kif5aa* mutant tectum. D = dorsal, V = ventral, R = rostral, C = caudal. (**B**) Quantification of total branch length of transplanted RGC axons at 5 and 7 dpf. Kif5aa mutant cell arbors are significantly smaller than wild-type cell arbors when growing into a wild-type tectum (p < 0.01). Wild-type cells built larger arbors when growing into a *kif5aa* mutant tectum (p < 0.05) (5 dpf: n = 14, 35, 6; 7 dpf: n = 14, 19, 5). (**C**) Schematic illustrating growth behavior of RGC axons in wildtype (upper panel) and kif5aa mutant tecta (lower panel) and upon loss of TrkC signaling (middle panel). Wild-type RGCs start to grow into the wild-type neuropil at 3 dpf. They grow highly active filopodial protrusions and start to form complex axonal arbors. At 5 dpf they reach their final size and maintain their branch shape at 7 dpf. When TrkC signaling is blocked by overexpression of a dominant negative receptor (ntrk3adN-GFP), wild-type cells show a substantially reduced arbor complexity (middle panel). Kif5aa mutant RGC arbors show a delay of ingrowth into the tectal neuropil. This is followed by a period of highly active growth with abundant filopodia formation. This results in highly complex arbors at 7 dpf. The delay of RGC growth is cell autonomous (***Figure 8—figure supplement 1***). The lack of retinal input leads to an upregulation of ntf3 expression by tectal cells and constitutes a growth-promoting environment.

The following figure supplements are available for figure 8:

**Figure supplement 1**. Phenotype of transplanted RGC arbors at early stages of development.

**Figure supplement 2**. Transplantation of *kif5aa* mutant RGCs into a *blumenkohl* mutant acceptor leads to an increased growth compared to transplantation into a wild-type acceptor.

Synapse assembly (as assayed by transport and distribution of Synaptophysin-containing clusters) was not detectably affected by the *kif5aa* mutation, in agreement with previous experiments in mouse KO cells (***Xia et al., 2003***). The observed higher percentage of mobile vesicles likely reflects the highly active growth at days 4 and 5, as it was previously shown in RGCs that synaptic puncta stability increases with axon maturation (***Meyer and Smith, 2006***). Filopodial activity was previously described as a sign of immature, silent axons before the onset of presynaptic activity (***Ben Fredj et al., 2010***). This is in line with the observed failure of *kif5aa* mutant axons to transmit neuronal signals.

As a secondary, non-cell autonomous consequence of *kif5aa* disruption, we observed that Ntf3 was upregulated by the tectum. Overexpression of Ntf3 in wild-type tectum and blockade of its receptor Ntrk3 (TrkC) in RGCs demonstrated that this neurotrophin is both sufficient and necessary to alter branch dynamics in RGC axon arbors. Classical work in the optic tectum of *Xenopus laevis* showed that overexposure to BDNF leads to enlarged and more complex axonal arbors (***Marshak et al., 2007***). Here, we identified the related Ntf3 as the intrinsic, growth-promoting signal for RGC axons in zebrafish.

Ntf3 upregulation was observed not only in *kif5aa* mutant larvae but also in two other previously characterized mutant lines, *blumenkohl* and *lakritz*, and when directly silencing RGCs via BoTx expression. In all cases, this upregulation was correlated with RGC absence or dysfunction. In *lakritz*

zebrafish larvae, RGCs are absent; the functional connection between the retina and the tectum thus is eliminated (*Kay et al., 2001*). In *blumenkohl*, transmitter release is diminished (*Smear et al., 2007*). Similar to our results with *kif5aa*, *blumenkohl* RGC axons show an increased arbor size and complexity (*Smear et al., 2007*). The same retrograde signal may underlie the enlargement of retinal arbors following treatment with MK-801, a blocker of glutamate receptors of the NMDA-subtype (*Schmidt et al., 2004*), or in *macho* mutants, in which RGCs fail to generate action potentials (*Gnuegge et al., 2001*). In the case of *lakritz*, when single wild-type RGCs were transplanted into a *lakritz* host, their solitary axons formed larger and more complex arbors (*Gosse et al., 2008*). This result is reminiscent of the phenotype of wild-type RGCs growing in a *kif5aa* mutant larva, that is, when they are surrounded by inactive axons. Interestingly, Ntf3 upregulation alone as observed in *blu* mutants or in overexpression experiments had no significant effect on mitochondria distribution and short-lived filopodia dynamic. This suggests that these phenotypes are specific to *kif5aa−/−* RGCs and probably caused by direct axonal trafficking defects, and that they are not due to impaired synaptic activity.

Together, these data suggest a model in which a deficit in presynaptic activity enhances the production and release of Ntf3 by tectal neurons. Tectum-derived Ntf3, in turn, retrogradely stimulates axonal branching and, thus, the addition of presynaptic terminals (*Figure 7C*). Such a signal could be the core motif of a compensatory pathway that is triggered when synaptic drive deviates from some homeostatic setpoint (*Davis and Bezprozvanny, 2001*; *Burrone and Murthy, 2003*). Our analysis of a mutation in a motor protein has thus unmasked a potentially general structural plasticity mechanism that together with the well-known competition-based and Hebbian mechanisms shapes the retinotectal projection and determines the final axonal arbor size (*Ruthazer et al., 2003*; *Ruthazer and Cline, 2004*; *Hua et al., 2005*; *Uesaka et al., 2006*; *Schwartz et al., 2009*, *2011*; *Ben Fredj et al., 2010*; *Munz et al., 2014*).

## Materials and methods

### Ethics statements

All fish are housed in the fish facility of our laboratory, which was built according to the local animal welfare standards. All animal procedures were performed in accordance with French and European Union animal welfare guidelines.

### Fish lines

The following transgenic fish lines were used or generated: *Tg(UAS:RFP, cry:eGFP)* (*Auer et al., 2014*), *Tg(UAS:SypGFP)* (*Meyer and Smith, 2006*), *Tg(HuC:GCaMP5)* (*Ahrens et al., 2013*), *Tg (BGUG)* (*Xiao and Baier, 2007*), *Tg(Pou3f4:Gal4)* (*Xiao and Baier, 2007*), *Tg(gSA2AzGFF49A)* (*Muto et al., 2013*), *Tg(Shh:eGFP)* (*Neumann and Nuesslein-Volhard, 2000*), *Tg(pou4f3:mGFP)* (*Xiao et al., 2005*), *Tg(Isl2b:Gal4, cmlc2:eGFP)*, *Tg(UAS:BoTxLCB-GFP)* (see 'Materials and methods'). TALENs used to generate the *kif5aa* loss-of-function alleles were described previously (*Auer et al., 2014*). All mutant and transgenic lines used in this study are described in *Supplementary file 1*.

### Genotyping of the *kif5aa*[*162] mutant alleles

For genotyping the following primers were used (5′ to 3′): kif5aa_geno_fwd: GTTCACAGATTGT-GATGTCTGTG, kif5aa_geno_rev: TGGAGGATGGAGAAATGATGACA. After PCR amplification from genomic DNA the 400 bp long amplicon was digested with NcoI. The wild-type allele is digested into two fragments of 240 bp and 160 bp length, respectively. The mutant alleles are not digested and show a band at 387 bp or 390 bp.

### Molecular cloning

The pIsl2b:Gal4, cmlc2:eGFP construct was generated by a Gateway reaction (MultiSite Gateway Three-Fragment Vector Construction Kit, ThermoFisher Scientific, Waltham, MA) using the p5E-Isl2b (*Ben Fredj et al., 2010*), pME-Gal4, p3E-pA and the pDest-Tol2;cmlc2:eGFP (*Kwan et al., 2007*) vectors. We generated a p5E-4nrUAS vector with four non repetitive UAS sequences by digestion of the 4Xnr UAS:GFP vector (*Akitake et al., 2011*) and insertion of the 4nrUAS fragment into the p5E-10UAS vector (*Kwan et al., 2007*) after HindIII and AleI digestion. To obtain constructs with multiple UAS sequences, we generated a p5E-4nrUAS-tagRFPCaax-pA-4nrUAS vector. The tagRFPCaax sequence was amplified with primers listed in *Supplementary file 2* and inserted into the pME-MCS vector (*Kwan et al., 2007*) after BamHI/NotI digestion resulting in pME-tagRFPCaax. After a Gateway

reaction (MultiSite Gateway Three-Fragment Vector Construction Kit) using the p5E-4nrUAS, pME-tagRFPCaax, p3E-pA, and the pDest-Tol2; cmlc2:eGFP (*Kwan et al., 2007*) vectors, the p4nrUAS:tagRFPCaax-pA-Tol2;cmcl2:eGFP vector was digested with StuI and SnaBI. The 4nrUAS:tagRFPCaax-pA fragment was subsequentially inserted into the StuI digested and dephosphorylated p5E-4nrUAS vector to create a p5E-4nrUAS-tagRFPCaax-pA-4nrUAS vector. We generated a pME-SypGFP vector by digestion of 5× UAS:SypGFP (*Meyer and Smith, 2006*) with EcoRI and NotI and insertion into the pME-MCS plasmid. To obtain the p4nrUAS:tagRFPCaax-pA-4nrUAS:SypGFP-pA-Tol2;cmcl2:eGFP vector we performed a Gateway reaction using p5E-4nrUAS-tagRFPCaax-pA-4nrUAS, pME-SypGFP, p3E-pA and pDest-Tol2;cmlc2:eGFP. We generated a pME-PhbGFP and a pME-Phbmcherry vector by digestion of pClontecN1-PhbGFP and pClontecN1-Phbmcherry (a kind gift from Christian Wunder) (*Rajalingam et al., 2005*) with EcoRI and NotI and insertion into pME-MCS. To obtain the p4nrUAS:tagRFPCaax-pA-4nrUAS:PhbGFP-pA-Tol2;cmcl2:eGFP vector we performed a Gateway reaction using p5E-4nrUAS-tagRFPCaax-pA-4nrUAS, pME-PhbGFP, p3E-pA and pDest-Tol2;cmlc2:eGFP. We generated a p5E-4nrUAS-SypGFP-pA-4nrUAS vector by performing a Gateway reaction using p5E-4nrUAS, pME-SypGFP, p3E-pA and pDest-Tol2; cmlc2:eGFP. The resulting p4nrUAS:SypGFP-pA-Tol2;cmcl2:eGFP vector was digested with StuI and SnaBI to create a p5E-4nrUAS-SypGFP-pA-4nrUAS vector after insertion into the StuI digested and dephosphorylated p5E-4nrUAS vector fragment. To create a p4nrUAS:SypGFP-pA-4nrUAS:PhBmcherry-pA-Tol2;cmcl2:eGFP plasmid we performed a Gateway reaction using p5E-4nrUAS-SypGFP-pA-4nrUAS, pME-Phbmcherry, p3E-pA and pDest-Tol2;cmcl2:eGFP. To create a ntf3_E2A_tagRFP expression construct we amplified ntf3_E2A from wild-type zebrafish cDNA (3 dpf) and fused it to a tagRFP fragment. Primers used are listed in *Supplementary file 2*. After digestion and insertion into the pME-MCS vector with HindIII and NotI, we performed a Gateway reaction using p5E-10UAS (*Kwan et al., 2007*), pME-ntf3_E2A_tagRFP, p3E-pA and pDest-Tol2; cmlc2:eGFP to generate p10UAS-ntf3_E2A_tagRFP-pA-tol2, cmcl2 :eGFP. To create a dominant negative ntrk3A expression construct, we amplified a truncated fragment of ntrk3A from wild-type zebrafish cDNA (3 dpf) and fused it to the eGFP open reading frame. Primers used are listed in *Supplementary file 2*. After digestion and insertion into the pME-MCS vector with HindIII and NotI we performed a Gateway reaction using p5E-10UAS, pME-ntrk3AdNeGFP, p3E-pA and pDest-Tol2; cmlc2:eGFP to generate p10UAS:ntrk3adNeGFP-pA-Tol2; cmcl2:eGFP. To create a pIsl2b:eGFPCaax construct, we performed a Gateway reaction using p5E-Isl2b, pME-eGFPCaax (*Kwan et al., 2007*), p3E-pA and pDest-Tol2; cmlc2:eGFP. To generate a UAS:BoTxBLC-GFP construct, a codon-optimized cDNA encoding botulinum toxin light chain B serotype (*Kurazono et al., 1992*; *Whelan et al., 1992*) was fused in frame with GFP and cloned downstream of the 5× UAS sequence using gateway recombination (*Asakawa and Kawakami, 2008*). Microinjection of the pT2UAS:BoTxBLC-GFP plasmid (50 ng/µl) was based on standard protocols with Tol2 mRNA (25 ng/µl). Over 50 founders were screened for the presence of a functional transgene using a combination of behavioral assays (touch-evoked swimming, escape response) and the level of expression of the BoTxBLC-GFP fusion protein (Suster et al., in preparation).

## Genetic linkage mapping of the *vertigo* mutation

The genomic locus of the *vertigo^s1614* allele (*vrt*) was determined using a PCR based simple sequence length polymorphisms (SSLPs) marker strategy. *Vrt* carriers in the TL genomic background (in which the mutagenesis was carried out) were crossed to the WIK genomic background to generate mapping crosses. From 1800 meioses the two SSLP markers, fj61a10 and tsub1g3, located 0.1 cM apart were identified to flank the *vrt* locus. Both markers are placed on Contig 963 of the Sanger center BAC sequencing project built by two overlapped BACs with sequencing information and four genes, one of which is kif5aa, are predicted between the two mapping markers.

## Immunohistochemistry and in situ hybridization

Retinal sections of 5 dpf old embryos and whole mount embryos were stained using standard protocols (*Kay et al., 2001*). The full list of primary and secondary antibodies is given below. Whole-mount in situ hybridization was performed according to standard protocols (*Di Donato et al., 2013*). *Kif5aa*, *ntrk3a*, *ntrk3b*, and *ntf3* specific sense and antisense probes were amplified by PCR from cDNA and cloned into the pCRII-topo vector (ThermoFisher Scientific). All primers used are reported in *Supplementary file 2*. The *tag-1* probe was synthesized from a 3.1 kb tag1 cDNA clone

(*Warren et al., 1999*) and the *pax2.1* probe was synthesized using the complete pax2.1 cDNA (*Krauss et al., 1991*). Probes were hydrolyzed to 200 bp fragments prior to use.

## Antibodies

The following primary antibodies were used in the course of this study: anti-Parvalbumine (EMD Millipore, Billerica, MA, MAB1572, 1:500), anti-eGFP (Genetex, GXT13970, 1:500), anti-PKC (Santa Cruz Biotechnology, Santa Cruz, CA, sc-209, 1:500), anti-humanNT3 (ThermoFisher Scientific, PA1-18385, 1:500), anti-alpha-tubulin (Genetex, Irvine, CA, GTX11304, 1:5000). The following secondary antibodies were used in the course of this study: anti-mouse-Alexa635 (ThermoFisher Scientific, A31574, 1:250), anti-rabbit-Alexa546 (ThermoFisher Scientific, A11081, 1:250), anti-chicken-Alexa488 (ThermoFisher Scientific, A11039, 1:250), anti-Rabbit IgG, HRP conjugated (Promega, W4011, 1:2000), anti-mouse IgG, HRP conjugated (Promega, Madison, WI, W4021, 1:2000).

## Optokinetic response

The behavioral test for the optokinetic response was performed as described previously (*Muto et al., 2005*).

## Single cell labeling and filopodia analysis

The morphology of single RGCs was analyzed using the *Tg(BGUG)*, *kif5aa*[*162]+/− transgenic line and single cells were imaged over consecutive days. To quantify filopodia dynamics, imaging was performed for 10 min at a rate of 1 frame/2 min. All branches not extending within this imaging period were assigned as stable branches and used for quantification of branch number and length. All branches extended or retracted within this imaging period were defined as filopodia. Single cell labeling to analyze synapse and mitochondria distribution was achieved by injection of 1 nl of naked plasmid DNA (25 ng/μl) into 1 cell stage embryos of the *Tg(Isl2b:Gal4, cmlc2:eGFP)*, *kif5aa*[*162]+/− transgenic line. The following constructs were used: p4nrUAS:tagRFPCaax-pA-4nrUAS:PhBGFP-pA-Tol2; p4nrUAS:tagRFPCaax-pA-4nrUAS:SypGFP-pA-Tol2;cmcl2:eGFP; p4nrUAS:SypGFP-pA-4nrUAS:PhBmcherry-pA-Tol2;cmcl2:eGFP. To generate single RGCs expressing the dominant negative ntrk3a receptor, we injected 1 nl of naked p10UAS:ntrk3aDN-eGFP-pA-Tol2;cmcl2:eGFP plasmid DNA into 1 cell stage embryos of the *Tg(Pou3f4:Gal4)*, nacre+/− transgenic line. To generate single eGFP expressing RGCs growing into a ntf3 overexpressing tectum, we performed injections of 0.1 ng/μl pIsl2b:eGFPCaax plasmid DNA, 15 ng/μl p10UAS:ntf3-E2A-tagRFP-pA-tol2, cmcl2:eGFP plasmid DNA and 50 ng/μl Tol2 transposase mRNA into 1 cell stage embryos of the *Tg(gSA2AzGFF49A)* (*Muto et al., 2013*) transgenic line. In control injections, no p10UAS:ntf3-E2A-tagRFP-pA-tol2, cmcl2:eGFP plasmid DNA was injected.

## Confocal microscopy

Imaging was performed on a Roper confocal spinning disk head mounted on a Zeiss upright microscope, and acquisitions were done with a CoolSNAP HQ2 CDD camera (Photometrics, USA) through the MetaMorph software (Molecular Devices, Sunnyvale, CA). Embryos were anaesthetized using 0.02% tricaine (MS-222, Sigma-Aldrich, Saint Louis, MO) diluted in egg water and embedded in 1% low melting-point agarose in glass-bottom cell tissue culture dish (Fluorodish, World Precision Instruments, Sarasota, FL). Acquisitions were done using water immersion long working distance lenses, at 40× magnification (W DIC PL APO VIS-IR; 421462-9900) for z-stack images of the whole tectum and at 63× magnification (W PL APO VIS-IR; 421480-9900) for single plane time-lapse imaging of linear axonal segments. Images were assembled and analyzed in ImageJ (NIH). Z-stack images were manual edited to remove skin autofluorescence.

## Time-lapse imaging, kymogram production and analysis

Time-lapse parameters were determined similar to previous studies (*Moughamian et al., 2013*; *Niwa et al., 2013*) based on the speed of transport in the tectum and set at 5 s intervals for 15 min (SypGFP) and 20 min (mitoGFP) total duration. Time-lapse images were assembled and analyzed in ImageJ to determine the percentage of moving vs stable particles, as well as distribution/density and size of the organelles. Kymograms were extracted for each linear segment using the kymogram tool (Montpellier RIO Imaging, CNRS, France). Extraction of small structures of the kymograms was done using the rotational watershed algorithm of the KymoMaker program (*Chiba et al., 2014*) and trace detection was done manually in ImageJ.

## Calcium imaging

5- to 7-day-old nacre (mitfa−/−) or TL larvae were taken from a cross of kif5aa*162+/− Tg(HuC:GCaMP5G) or Tg(lsl2b:Gal4), Tg(UAS:GCaMP3) transgenic fish. They were immobilized in 2% low melting point agarose and mounted with the dorsal side up on a plexiglas platform. The platform was then placed in a custom-made chamber and immersed in E3 solution without methylene blue. The agarose around the eyes was cut away with a scalpel to allow for an unhindered view for the larvae. The larvae faced with one eye towards a glass cover slip in the chamber wall at a distance between 8 and 10 mm. Directly behind this glass cover slip and outside the chamber, a monochromatic OLED array (800 × 600 px, 13 × 9 mm, eMagin) for visual stimulus presentation covering approximately 70° by 50° of the larva's visual field was positioned. A colored filter (Kodak Wratten No. 32) was placed between glass cover slip and OLED to block green light emitted from the OLED thus allowing for simultaneous imaging and visual stimulation. Visual stimuli were synchronized to the acquisition and consisted of single black bars on a white background (or the inverse) running along the caudal rostral axis. The long axis of the bar was orthogonal to the direction of motion. Each bar was approximately 7° in width and moved at 16°/s. Visual stimuli were generated and controlled by custom scripts written in Matlab (MathWorks, for details see Source code 1) using the Psychophysics Toolbox extensions (Brainard, 1997; Pelli, 1997). Confocal imaging of visually-evoked calcium responses in the tectum contralateral to the eye receiving the visual stimulus was performed using an upright microscope (Roper/Zeiss, Germany) equipped with a Spinning Disk head (CSU-X1, Yokogawa, Japan) and a 40×/1.0 NA water-immersion objective (Zeiss). Time-series streams of 5 min duration were acquired at 4 Hz with 0.323 × 0.323 µm spatial resolution (620 × 520 pixels). GCaMP3 or GCaMP5G were excited by a 491 nm laser and emitted light was bandpass-filtered (HQ 525/50). Images were taken with a CCD camera (CoolSnap HQ2 Photometrics, Tucson, AZ). We did not observe any differences in larvae from 5 dpf to 7 dpf. Therefore, the data sets were combined for analysis. Occasionally, a visual response at the onset of the laser illumination or the power-on of the OLED was observed that quickly returned to baseline, probably due to habituation of the fish. Therefore, we excluded the first few seconds from acquisition analysis. Each stimulus epoch was presented for 4.4 s to every animal with an inter-epoch interval of 5.6 or 10.6 s to allow for the GCaMP signal to return to baseline values. After the experiment, fish were genotyped. Confocal time-series were pre-processed by correcting for motion with a translation algorithm (Fiji [Schindelin et al., 2012]). For each acquisition, ROIs for Neuropil and PVNs, respectively, were determined manually. Then the averaged ROI based time-series were smoothened by a low pass filter and the baseline signal was calculated by calculating the minimum in a time interval of 10 s before stimulus onset. The smoothened fluorescence signal and the baseline fluorescence were then used to calculate normalized signal intensity changes (% ΔF/F0). To identify pixels that were response locked to the stimulus, we performed linear regression on the Calcium evoked time-series (Miri et al., 2011). For this, we convolved the stimulus time-series with an exponentially decaying kernel with half-decay times for GCaMP5G (667 ms) (Akerboom et al., 2012) or GCaMP3 (597 ms) (Tian et al., 2009). This predicted fluorescence trace for the bar stimulus was then compared with the measured calcium traces for each pixel using Pearson correlation.

## Transmission electron microscopy

6 dpf larvae were anaesthesized in 0.004% Tricaine in E3 solution and then fixed in 2% glutaraldehyde in 0.1 M phosphate buffer, pH 7.2. Tails were severed to increase permeability of fixation. Four drops of 4% OsO4 were added to 1 ml of glutaraldehyde fixative, and samples were soaked for 15 min. After three washes in 0.1 M phosphate buffer, samples were stored in 2% glutaraldehyde solution. Prior to embedding, samples were washed three times with 0.1 M phosphate buffer for 2 min and dehydrated with graded acetone: 35% acetone, 50% acetone, 75% acetone, 80% acetone, 95% acetone, and 100% acetone (three times) for 10 min per solution while shaking. Samples were infiltrated with Epon resin/acetone mixtures: Epon resin: acetone (1:3), Epon resin: acetone (1:1) and Epon resin: acetone (3:1) followed by pure Epon resin containing the accelerant BDMA (three washes). Finally, embedded samples were cured in a vacuum oven and sectioned with a RMC MT6000 Microtome to 70 nm slices. Images were acquired with an FEI Tecnai 12 Transmission electron microscope.

## Quantitative RT-PCR

Total RNA was prepared from 4 or 5 dpf embryos with TRIzol reagent (ThermoFisher Scientific) and TURBO DNA-free reagents (ThermoFisher Scientific). RNA (1 µg) was retrotranscribed using random

primers and the SuperScript III First-Strand Synthesis system (ThermoFisher Scientific). For q-RT-PCR, the SYBR Green PCR Master Mix (ThermoFisher Scientific) was used according to the manufacturers protocol and the PCR reaction was performed on an ABI PRISM 7900HT instrument. *Ef1a* and *RPL13a* were used as reference genes as reported previously (*Tang et al., 2007*). All assays were performed in triplicate using 11.25 ng of cDNA per reaction. The mean values of triplicate experiments were calculated according to the delta*CT* quantification method.

## Western blotting

Western blot analysis of embryo extracts was performed using standard techniques. Briefly, about 25 5 dpf larvae were homogenized in lysis buffer (20 μl/embryo) containing: 10 mM HEPES, 300 mM KCl, 5 mM $MgCl_2$, 0.45% Triton, 0.05% Tween, Protease inhibitor-EDTA (Mini Complete, Roche, Switzerland). Protein extracts (about 20 μg/lane) were separated by SDS-PAGE and subsequently blotted onto a PVDF membrane. Secondary antibody couples with Horseradish peroxidase (1:2000) were used to detect the anti-NTF3 (1:500) and anti-alpha-Tubulin (1:5000) primary antibodies (see Antibody section above for details), and reveled using ECL Western Blotting Detection Reagents (GE Healthcare Life Sciences, Pittsburgh, PA). Western blot quantification was performed using a cheminoluminescence digital imaging system (ImageQuant Las-4000 Mini, GE Healthcare Life Sciences) and analyzed using ImageJ software.

## Acknowledgements

We want to thank F Engert for the *Tg(HuC:GCaMP5)* transgenic line and C Nuesslein-Vollhard for the *Tg(shh:eGFP)* transgenic line. Furthermore, we are grateful to M Meyer for sharing the 5xUAS:SypGFP construct and C Wunder for sharing the PhB:eGFP and PhB:mcherry constructs. We thank Manuela Portoso for helpful advice with the Western Blot analysis and all members of the Del Bene lab for fruitful discussions. We thank the Developmental Biology Curie imaging facility (PICT-IBiSA@BDD, Paris, France, UMR3215/U934) member of the France-BioImaging national research infrastructure for their help and advice with confocal microscopy. The Del Bene laboratory 'Neural Circuits Development' is part of the Laboratoire d'Excellence (LABEX) entitled DEEP (ANR-11-LABX-0044), and of the École des Neurosciences de Paris Ile-de-France network. TOA was supported by a Boehringer Ingelheim Fonds Ph.D. fellowship and VB by an FRSQ and CIHR Doctoral Award and is enrolled in the ENP Graduate Program. CG was supported by a FRM postdoctoral fellowship. This work has been supported by an ATIP/AVENIR program starting grant (FDB), ERC-StG #311159 (FDB), ANR-II-INBS-0014 (JPC), CNRS, INSERM and Institut Curie core funding.

## Additional information

### Funding

| Funder | Grant reference | Author |
| --- | --- | --- |
| ATIP/Avenir starting grant CNRS/INSERM | Starting Grant | Filippo Del Bene |
| European Research Council (ERC) | ERC-StG#311159 | Filippo Del Bene |
| Boehringer Ingelheim Fonds | Graduate Student Fellowship | Thomas O Auer |
| Fonds de Recherche du Québec - Santé | Graduate Student Fellowship | Valerie Bercier |
| Fondation pour la Recherche Médicale | Postdoctoral Fellowship | Christoph Gebhardt |
| National Funding Agency for Research (ANR) | ANR-II-INBS-0014 | Jean-Paul Concordet |

The funders had no role in study design, data collection and interpretation, or the decision to submit the work for publication.

## Author contributions
TOA, Conception and design, Acquisition of data, Analysis and interpretation of data, Drafting or revising the article; TX, Conception and design, Acquisition of data, Analysis and interpretation of data; VB, CG, KD, Acquisition of data, Analysis and interpretation of data; J-PC, MS, KK, Drafting or revising the article, Contributed unpublished essential data or reagents; CW, Conception and design, Contributed unpublished essential data or reagents; JW, Conception and design, Drafting or revising the article; HB, FDB, Conception and design, Analysis and interpretation of data, Drafting or revising the article

## Author ORCIDs
Joachim Wittbrodt, http://orcid.org/0000-0001-8550-7377
Filippo Del Bene, http://orcid.org/0000-0001-8551-2846

## Ethics
Animal experimentation: All fish are housed in the fish facility of our laboratory, which was built according to the local animal welfare standards. All animal procedures were performed in accordance with French and European Union animal welfare guidelines.

## Additional files

### Supplementary files
• Supplementary file 1. Description of zebrafish mutant and transgenic lines used in this study.

• Supplementary file 2. Primers used in this study.

• Source code 1. Source code for moving bar stimulus.

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
