## [Decision Letter]

Thank you for sending your work entitled “Deletion of a kinesin I motor unmasks a mechanism of homeostatic branching control by neurotrophin-3” for consideration at *eLife*. Your article has been evaluated by a Senior editor, a Reviewing editor and two reviewers. The Reviewing editor and the reviewers discussed their comments before we reached this decision, and the Reviewing editor has assembled the following comments to help you prepare a revised submission.

In summary, there is high enthusiasm for both the subject matter and the potential nature of the advance. All agree that the manuscript would be of broad interest to the readership of *eLife*. However, there is consensus among the reviewers that some of the major conclusions of the work are not yet supported sufficiently. Please see the reviewer comments below, which have been organized into major issues that must be addressed and minor issues that can be addressed at your discretion.

In the present article, Auer et al. present a set of results suggesting that (1) the Kinesin 1-family anterograde microtubule-based motor Kif5 is required for proper axon morphogenesis, (2) in *kif5aa* mutant zebrafish, retino-tectal axons display delayed target invasion but ultimately increased axon branching and (3) reduced anterograde axonal transport of mitochondria resulting in depletion of mitochondria from the distal part of the axon and (4) upregulation of NT3 in the tectum of *kif5aa* mutants. The results corresponding to the first 3 parts are fairly convincing. The topic is interesting and the study could provide new insights into the molecular mechanisms underlying axon morphogenesis.

Major Criticisms:

1) The final part of the manuscript suggesting that NT3 upregulation in the tectum is responsible for the overgrowth of axons observed in the *kif5aa* mutant is not convincing and the lack of causality between the two phenotypes is weakening the main conclusions. Specifically, the authors do not provide any evidence for a causal relationship between depletion of mitochondria from distal axon and (1) delayed axon invasion and/or (2) increased axon branching. In fact, since the authors suggest later on in the article that the increased axon branching might be largely the results of cell non-autonomous effect of NT3 upregulation by tectal cells, it becomes important to test if the axon growth/branching phenotype is causally related with the mitochondria transport defect and/or the NT3 upregulation.

Could the authors also show the quantification of filopodia increase after neurotrophic Factor 3 overexpression? Is there an additional effect when Kif5aa is deleted and neurotrophic factor 3 is overexpressed, or is the branching/filopodia changed to a similar extend compared to either Kif5aa deletion in retinal neurons or overexpression of neurotrophic factor 3 in tectal/glial cells? Likewise, if the authors overexpressed the dominant negative *ntrk3a* (TrkC receptor homologue) kinase dead version in *kif5aa* mutant animals, would there be a reduction in branching? How is the branching compared to wildtype animals?

2) Please overexpress neurotrophic Factor 3 in the tectal/glial cells and express the dominant negative *ntrk3a* mutant in retinal cells.

3) The authors should provide other evidence besides loss of Kif5aa that loss of activity in these RGCs leads to increased Ntf3 expression the tectum, such as tetanus toxin expression in RGC to silence neuronal activity in a cell-autonomous manner.

4) *Kif5aa* mutant fish die at 10dpf (lack of swim bladder inflation) making it impossible to address the possibility that Ntf3 upregulation may eventually rescue the reduced mitochondrial anterograde transport phenotype. However, the *blumenkohl* mutant, which is viable and has increased axonal branching but decreased visual transmission, could be used to determine if at a later stage increased Ntf3 in the tectum leads to increased mitochondrial localization in the RGC axons.

5) The authors show/discuss that mitochondria are often associated with presynaptic boutons (Obashi and Okabe EJN 2013; Courchet, Lewis et al, Cell 2013). They demonstrate that they can perform dual channel imaging of mitochondria and stable presynaptic sites (Figure 5—figure supplement 1). The authors should address if the fraction of mitochondria associated with presynaptic boutons is affected upon loss of Ki5aa and if Ntf3 overexpression can rescue/affect this localization to help link the two stories together.

6) Another way to potentially link the two parts of the paper would to test if reduction of NT3 expression in the tectum of *kif5aa* mutants normalizes the axon branching phenotype.

Minor comments:

The authors start out in the Abstract about how important microtubules are to regulate neuronal polarization also during, but then nothing about these aspects are found in the Introduction. Here, it would be worthwhile to spend a few additional words about how microtubules drive the development of neuronal polarization (e.g. Witte et al., JCB, 2008; Gomis-Rüth et al., Current Biology, 2008).

[Editors' note: further revisions were requested prior to acceptance, as described below.]

Thank you for resubmitting your work entitled “Deletion of a kinesin I motor unmasks a mechanism of homeostatic branching control by neurotrophin-3” for further consideration at *eLife*. Your revised article has been favorably evaluated by a Senior editor and a member of the Board of Reviewing Editors. The manuscript has been improved but there is one remaining issue that needs to be addressed before acceptance, as outlined below:

The additional work undertaken in this revised version of the paper utilises a new strain of fish: Tg(UAS:BoTxLCB-GFP). In order to provide as complete a picture as possible we ask that you please include details of the development of this reagent.

---

## [Author Response]

*1) The final part of the manuscript suggesting that NT3 upregulation in the tectum is responsible for the overgrowth of axons observed in the* kif5aa *mutant is not convincing and the lack of causality between the two phenotypes is weakening the main conclusions. Specifically, the authors do not provide any evidence for a causal relationship between depletion of mitochondria from distal axon and (1) delayed axon invasion and/or (2) increased axon branching. In fact, since the authors suggest later on in the article that the increased axon branching might be largely the results of cell non-autonomous effect of NT3 upregulation by tectal cells, it becomes important to test if the axon growth/branching phenotype is causally related with the mitochondria transport defect and/or the NT3 upregulation*.

To determine if the observed increase in axon growth and branching was directly linked to the mitochondria transport defects, we analyzed mitochondria distribution in *blu* mutants, lacking synaptic transmission between RGCs and the tectum and showing upregulation of Ntf3. In these animals although RGCs show increased arbor size and we detect increased Ntf3 expression, no change in mitochondria localization was observed both by in vivo imaging and EM analysis. In contrast our data show that an upregulation of Ntf3 via overexpression in the tectum alone is sufficient to induce overgrowth in wildtype RGCs. These data are now presented in Figure 5—figure supplement 2 and Figure 5—figure supplement 3 and discussed in the text, as follows:

“Mitochondria are preferentially localized at active synapses (68) and are found in close proximity to stable Synaptophysin-containing clusters in RGC arbors (Figure 5—figure supplement 2). By co-labelling mitochondria and presynaptic clusters in the same cells in vivo we showed that approximately 40% stable presynaptic sites are associated with mitochondria in wildtype and *blu-/-* RGC axons (Figure 5—figure supplement 2).”

“In addition we observed that the distribution of mitochondria was not significantly altered in *blu-/-* RGC axons nor was the association with stable presynaptic sites (Figure 5—figure supplement 2 and Figure 5—figure supplement 3). Together these data suggest that Ntf3 upregulation does not per se affect mitochondria localization.”

“Interestingly, Ntf3 upregulation alone as observed in *blu* mutants or in overexpression experiments had no significant effect on mitochondria distribution and short-lived filopodia dynamic. This suggests that these phenotypes are specific to *kif5aa-/-* RGCs and probably caused by direct axonal trafficking defects, and that they are not due to impaired synaptic activity.”

*Could the authors also show the quantification of filopodia increase after neurotrophic Factor 3 overexpression? Is there an additional effect when Kif5aa is deleted and neurotrophic factor 3 is overexpressed, or is the branching/filopodia changed to a similar extend compared to either Kif5aa deletion in retinal neurons or overexpression of neurotrophic factor 3 in tectal/glial cells? Likewise, if the authors overexpressed the dominant negative* ntrk3a *(TrkC receptor homologue) kinase dead version in* kif5aa *mutant animals, would there be a reduction in branching? How is the branching compared to wildtype animals*?

We have now extended the analysis of filopodia dynamics via short term time lapse imaging, to *blu-/-* RGC axons and wildtype axons growing in tecta overexpressing Ntf3. In both cases filopodia dynamics were comparable to what is seen in wildtype cells and reduced when compared to *kif5aa*-/- mutant cells. These new data suggest that the overgrowth phenotype and the increased filopodia dynamics observed in *kif5aa-/-* are two independent phenotypes. Therefore Ntf3 upregulation can cause RGCs overgrowth without affecting filopodia dynamics. These new data are now presented in Figure 7—figure supplement 1 and in the main text:

“We next decided to test if the observed increased formation and retraction of filopodia in the axons of *kif5aa* mutant RGCs was directly caused by Ntf3 upregulation. Therefore we analyzed filopodia dynamics via time-lapse imaging both in *blu-/-* RGCs and in axons growing when Ntf3 was overexpressed in the tectum via our transgenic construct. In both experimental conditions we did not observe any increase in the filopodia dynamics or RGC axons (Figure 7—figure supplement 1), excluding the possibility that Ntf3 overexpression has a direct effect on this process.”

“Interestingly, Ntf3 upregulation alone as observed in *blu* mutants […] not due to impaired synaptic activity.”

As the reviewers suggested we have analyzed axonal branching upon overexpression of the dominant negative *ntrk3a* in *kif5aa-/-* RGCs and observed a loss of the overgrowth phenotype at later stages and a reduced branch number when compared to wildtype RGCs. This reduced branching was also visible in wildtype RGCs overexpressing the dominant negative *ntrk3a* construct, suggesting a role for this receptor in branching behavior (Figure 7). Therefore, the text now reads:

“Upon overexpression of the truncated *ntrk3adN-GFP* construct in single wildtype RGCs by mosaic DNA expression, we observed a substantial reduction of axon branch length and number of branches at 5dpf (Figure 7), consistent with a role of Ntrk3 as a branch-promoting receptor.

In addition, *ntrk3adN-GFP* overexpression in single *kif5aa-/-* and *blu-/-* RGCs could abolish the axonal overgrowth normally observed in these mutants as measured by total branch length (85) and even reduce the number of branches compared to wildtype cells (Figure 7).”

*2) Please overexpress neurotrophic Factor 3 in the tectal/glial cells and express the dominant negative* ntrk3a *mutant in retinal cells*.

This is a great suggestion but unfortunately this experiment is currently technically not feasible. In our paper both Ntf3 and Ntrk3adN-GFP are overexpressed robustly using the Gal4/UAS system and different Gal4 lines specific for the tectum or the RGCs respectively. Combing them for the purpose of this experiment would result in an overexpression of both constructs, in both cell types. However, to try to circumvent this crossover, we generated a construct that drives the expression of Ntrk3adN-GFP directly under the control of the RGC specific promoter *islet2b* and injected it in embryos overexpressing Ntf3 in the tectum (with the GAL4/UAS system). Unfortunately the Ntrk3adN-GFP expression was not high enough to achieve a dominant negative effect per se and the fluorescence was so weak that it was not detectable anymore at 5dpf. We then tried to address this point indirectly by overexpressing Ntrk3adN-GFP in single RGCs of *blu-/-* mutants, where Ntf3 is also upregulated in the tectum. This experiment showed that interference with Ntf3 signaling was able to rescue RGC growth, and even produced arbors that were less complex, with reduced number of branches as compared to wildtype RGCs. These new data are now shown in Figure 7 and included in the text:

“In addition, *ntrk3adN-GFP* overexpression in single *kif5aa-/-* and *blu-/-* RGCs could abolish the axonal overgrowth normally observed in these mutants as measured by total branch length (85) and even reduce the number of branches compared to wildtype cells (Figure 7).”

*3) The authors should provide other evidence besides loss of Kif5aa that loss of activity in these RGCs leads to increased Ntf3 expression the tectum, such as tetanus toxin expression in RGC to silence neuronal activity in a cell-autonomous manner*.

We agree that this point is crucial for our manuscript. However, it is important to consider that the existing tetanus toxin transgenic lines in zebrafish are not expressed in a sufficiently strong way to address it. We therefore asked our collaborators Maximiliano Suster, Claire Wyart and Koichi Kawakami to share with us a new unpublished transgenic line expressing botulinum toxin light chain B under the control of the UAS promoter (they have been added as co-authors in this revised manuscript). Their experiments (soon to be published) have demonstrated the superior ability of this line to silence neuron activity when compared to TeTx. Using this tool we showed via qRT-PCR and Western blot analysis, that complete silencing of RGCs indeed cause Ntf3 overexpression. These data are shown in Figure 6—figure supplement 1, and included in the main text:

“To directly test the causality link between the lack of presynaptic input and upregulation of Ntf3 in the optic tectum we silenced the neuronal activity of RGCs […]. These results strongly suggest that lack of presynaptic activity and subsequent overexpression of Ntf3 in the tectum trigger the increased size of axonal branches of RGCs in *kif5aa*, *blumenkohl* and *lakritz* mutants (see Figure 2; [33]; [85]).”

*4)* Kif5aa *mutant fish die at 10dpf (lack of swim bladder inflation) making it impossible to address the possibility that Ntf3 upregulation may eventually rescue the reduced mitochondrial anterograde transport phenotype. However, the* blumenkohl *mutant, which is viable and has increased axonal branching but decreased visual transmission, could be used to determine if at a later stage increased Ntf3 in the tectum leads to increased mitochondrial localization in the RGC axons*.

To address this possibility we have analyzed mitochondria distribution in *blu* mutant RGCs by in vivo imaging. No difference was observed at any stage as compared to the wildtype, although we could not extend our analysis after day 7 due to weak expression of the mito-GFP marker at later stages. Furthermore electron microscopy imaging showed no significant change in mitochondria density in *blu-/-* tecta. Interestingly in *blu-/-* mutant, where we observed Ntf3 overexpression and RCGs axons are expanded, the fraction of presynaptic clusters (visualized by synaptophysin-GFP) associated with mitochondria was unaffected, suggesting that Ntf3 does not have an impact on mitochondria association with synapses. These new data are now discussed in the Results and in the Discussion, and shown in Figure 5, and Figure 5—figure supplement 2 and Figure 5—figure supplement 3:

“Mitochondria are preferentially localized at active synapses (68) […]. 40% stable presynaptic sites are associated with mitochondria in wildtype and *blu-/-* RGC axons (Figure 5—figure supplement 2).”

“In addition we observed that the distribution of mitochondria was not significantly altered in *blu-/-* RGC axons […]. Ntf3 upregulation does not per se affect mitochondria localization.”

“Interestingly, Ntf3 upregulation alone as observed in *blu* mutants […] not due to impaired synaptic activity.”

*5) The authors show/discuss that mitochondria are often associated with presynaptic boutons (Obashi and Okabe EJN 2013; Courchet, Lewis et al, Cell 2013). They demonstrate that they can perform dual channel imaging of mitochondria and stable presynaptic sites (*Figure 5—figure supplement 1*). The authors should address if the fraction of mitochondria associated with presynaptic boutons is affected upon loss of Ki5aa and if Ntf3 overexpression can rescue/affect this localization to help link the two stories together*.

We carefully quantified the fraction of presynaptic puncta associated with mitochondria in wildtype RGCs and extended our analysis to trigeminal ganglion cells where the co-labeling of mitochondria and synaptic puncta is technically more feasible for later stages. In both cases we observed about 40% of presynaptic boutons associated with mitochondria. In *kif5aa-/-* trigeminal ganglion cells this fraction was significantly reduced to less than 20%. Unfortunately the mito-RFP transient labeling was not stable enough in *kif5aa-/-* RGCs to perform this analysis as we were afraid of not being able to distinguish between cells not expressing the RFP marker from cells depleted of mitochondria. Interestingly in *blu-/-* mutant, where we observe Ntf3 overexpression and RCGs axons are expanded, the fraction of presynaptic boutons associated with mitochondria was unaffected. This suggests that Ntf3 does not have a direct role in mitochondria localization. We have now added these results in the main text in the Results and Discussion and the corresponding images are shown in Figure 5—figure supplement 2.

*6) Another way to potentially link the two parts of the paper would to test if reduction of NT3 expression in the tectum of* kif5aa *mutants normalizes the axon branching phenotype*.

This experiment is an excellent suggestion. Indeed we tried to interfere with Ntf3 activity using Ntf3 antibody (as described by Williams JA et al. Dev Biol 2000) and by vivo-morpholino injection in the tectum (Kizil C, Brand M. PlosONE 2011). Unfortunately neither approach gave conclusive results. The antibody injection were performed in the brain ventricle or in the tectal neuropil, but had no lasting effect in our experiments when analyzing cells two days after the injection. In contrast, the vivo morpholino injection seemed to trigger a general toxic effect reducing axonal growth in an unspecific manner, as shown by the injection of control vivo morpholino. The best way to address this question would be to use a genetic tool designed to delete Ntf3 gene in a tissue specific manner, targeting only the optic tectum, but to our best knowledge, such a tool is unfortunately not available at the moment.

*Minor comments*:

*The authors start out in the Abstract about how important microtubules are to regulate neuronal polarization also during, but then nothing about these aspects are found in the Introduction. Here, it would be worthwhile to spend a few additional words about how microtubules drive the development of neuronal polarization (e.g. Witte et al., JCB, 2008; Gomis-Rüth et al., Current Biology, 2008)*.

Thanks for pointing out this inconsistency between our Abstract and Introduction. We have now expanded our Introduction adding the following sentences:

“Microtubules serve as main longitudinal cytoskeletal tracks in axons and it is well established that microtubule stabilization is a landmark of early axonal development that is sufficient to induce axon formation in vivo. Besides, microtubule stabilization alone can even lead to the transformation of mature dendrites into axons in differentiated neurons (32; 100; 101).”

[Editors' note: further revisions were requested prior to acceptance, as described below.]

*Thank you for resubmitting your work entitled “Deletion of a kinesin I motor unmasks a mechanism of homeostatic branching control by neurotrophin-3” for further consideration at* eLife*. Your revised article has been favorably evaluated by a Senior editor and a member of the Board of Reviewing Editors. The manuscript has been improved but there is one remaining issue that needs to be addressed before acceptance, as outlined below*:

*The additional work undertaken in this revised version of the paper utilises a new strain of fish: Tg(UAS:BoTxLCB-GFP). In order to provide as complete a picture as possible we ask that you please include details of the development of this reagent*.

We have now added the following detailed description regarding the generation of this reagent in the Materials and methods section.

To generate a UAS:BoTxBLC-GFP construct, a codon-optimized cDNA encoding botulinum toxin light chain B serotype (44; 99) was fused in frame with GFP and cloned downstream of the 5x UAS sequence using gateway recombination (6). Microinjection of the pT2UAS:BoTxBLC-GFP plasmid (50 ng/µl) was based on standard protocols with Tol2 mRNA (25 ng/µl). Over 50 founders were screened for the presence of a functional transgene using a combination of behavioral assays (touch-evoked swimming, escape response) and the level of expression of the BoTxBLC-GFP fusion protein.